# Functional interrogation of HOXA9 regulome in MLLr leukemia via reporter-based CRISPR/Cas9 screen

Hao Zhang[1,2†], Yang Zhang[3,4†], Xinyue Zhou[1,2†], Shaela Wright[3,4], Judith Hyle[3,4], Lianzhong Zhao[1,2], Jie An[1,2], Xujie Zhao[5], Ying Shao[6], Beisi Xu[6], Hyeong-Min Lee[7], Taosheng Chen[7], Yang Zhou[8], Xiang Chen[6], Rui Lu[1,2*], Chunliang Li[3,4*]

[1]Division of Hematology/Oncology, University of Alabama at Birmingham, Birmingham, United States; [2]O'Neal Comprehensive Cancer Center, University of Alabama at Birmingham, Birmingham, United States; [3]Department of Tumor Cell Biology, St. Jude Children's Research Hospital, Memphis, United States; [4]Cancer Biology Program/Comprehensive Cancer Center, St. Jude Children's Research Hospital, Memphis, United States; [5]Department of Pharmaceutical Sciences, St. Jude Children's Research Hospital, Memphis, United States; [6]Department of Computational Biology, St. Jude Children's Research Hospital, Memphis, United States; [7]Department of Chemical Biology and Therapeutics, St. Jude Children's Research Hospital, Memphis, United States; [8]Department of Biomedical Engineering School of Engineering, University of Alabama at Birmingham, Birmingham, United States

*For correspondence:
ruilu1@uabmc.edu (RL);
chunliang.li@stjude.org (CL)

†These authors contributed equally to this work

Competing interests: The authors declare that no competing interests exist.

**Abstract** Aberrant *HOXA9* expression is a hallmark of most aggressive acute leukemias, notably those with KMT2A (MLL) gene rearrangements. *HOXA9* overexpression not only predicts poor diagnosis and outcome but also plays a critical role in leukemia transformation and maintenance. However, our current understanding of *HOXA9* regulation in leukemia is limited, hindering development of therapeutic strategies. Here, we generated the *HOXA9-mCherry* knock-in reporter cell lines to dissect *HOXA9* regulation. By utilizing the reporter and CRISPR/Cas9 screens, we identified transcription factors controlling *HOXA9* expression, including a novel regulator, USF2, whose depletion significantly down-regulated *HOXA9* expression and impaired MLLr leukemia cell proliferation. Ectopic expression of Hoxa9 rescued impaired leukemia cell proliferation upon USF2 loss. Cut and Run analysis revealed the direct occupancy of USF2 at *HOXA9* promoter in MLLr leukemia cells. Collectively, the *HOXA9* reporter facilitated the functional interrogation of the *HOXA9* regulome and has advanced our understanding of the molecular regulation network in *HOXA9*-driven leukemia.

## Introduction

Dysregulation of the homeobox (HOX)-containing transcription factor *HOXA9* is a prominent feature in most aggressive acute leukemias (*Collins and Hess, 2016a*; *Alharbi et al., 2013*). During normal hematopoiesis, HOXA9 plays a critical role in hematopoietic stem cell expansion and is epigenetically silenced during lineage differentiation (*Alharbi et al., 2013*). In certain leukemia subtypes, this regulatory switch fails and *HOXA9* is maintained at high levels to promote leukemogenesis. However, the mechanisms governing *HOXA9* expression remain to be fully understood. *HOXA9* overexpression is commonly observed in over 70% of human acute myeloid leukemia (AML) cases and ~10% of acute lymphoblastic leukemia (ALL) cases (*Jambon et al., 2019*). Notably, the high

expression of *HOXA9* is sharply correlated with poor prognosis and outcome in human leukemia (*Golub et al., 1999*; *Baccelli et al., 2019*). An accumulating body of evidence indicates that *HOXA9* dysregulation is both sufficient and necessary for leukemic transformation (*Collins and Hess, 2016a*; *Alharbi et al., 2013*). Forced expression of *HOXA9* enforces self-renewal, impairs myeloid differentiation of murine marrow progenitors, and ultimately leads to late onset of leukemia transformation (*Bach et al., 2010*), which is accelerated by co-expression with interacting partner protein MEIS1 (*Kroon, 1998*). Conversely, knocking down *HOXA9* expression results in leukemic cell differentiation and apoptosis (*Ayton and Cleary, 2003*; *Zeisig et al., 2004*). Thus, excessive *HOXA9* expression has emerged as a critical mechanism of leukemia transformation in many hematopoietic malignancies.

Consistent with the broad overexpression pattern of *HOXA9* in many leukemia cases, a wide variety of genetic alterations in leukemia contribute to *HOXA9* dysregulation including *MLL* gene rearrangements (MLLr), *NPM1* mutations, *NUP98*-fusions, *EZH2* loss-of-function mutations, *ASXL1* mutations, *MOZ* fusions and other chromosome alterations (*Collins and Hess, 2016a*; *Jambon et al., 2019*; *De Braekeleer et al., 2014*; *Collins and Hess, 2016b*). Additionally, our recent work shows that *DNMT3A* hotspot mutations may also contribute to *HOXA9* overexpression by preventing DNA methylation at its regulatory regions (*Lu et al., 2016*). Given that genomic variation of *HOXA9* including NUP98-HOXA9 fusion and gene amplification accounted for less than 2% of *HOXA9* overexpression in AML cases (*Xu et al., 2016*; *Gough et al., 2011*; *Nakamura et al., 1996*), uncovering the upstream epigenetic and transcriptional regulators of *HOXA9* in leukemia could advance the design of novel therapeutic interventions. For example, because MLLr proteins recruit the histone methyltransferase DOT1L to the *HOXA* locus promoting hyper-methylation at histone H3 lysine 79 and subsequent high *HOXA9* transcription (*Krivtsov et al., 2008*), selective DOT1L inhibitors have been exploited to inhibit leukemia development and *HOXA9* expression in MLLr leukemias and are now in clinical trials (*Chen et al., 2015*; *Stein and Tallman, 2015*). However, DOT1L inhibitors usually act slowly and their effects remain sub-optimal. To date, most known *HOXA9* regulator proteins are epigenetic modifiers, and little is known about which DNA-binding transcription factors are involved in directly regulating *HOXA9* expression in acute leukemia (*Godfrey et al., 2017*; *Daigle et al., 2011*; *Yu et al., 2012*; *Shi et al., 2012*).

Previous studies have also advocated that the organization of chromatin domains at the *HOXA* gene cluster contributes to high *HOXA9* expression in cancer cells (*Luo et al., 2018*; *Xu et al., 2014*). Specifically, CCCTC-binding factor CTCF may potentiate *HOXA9* expression through direct binding at the conserved motif between *HOXA7* and *HOXA9* (CBS7/9) to establish necessary chromatin looping interaction networks in MLLr AML MOLM13 cells (*Luo et al., 2018*; *Luo et al., 2019*). In contrast, Ghasemi et al. reported that *HOXA* gene expression was maintained in the CTCF-binding site deletion mutants derived from AML OCI-AML3 cells, suggesting that transcriptional activity at the *HOXA* locus in NPM1-mutant AML cells does not require long-range CTCF-mediated chromatin interactions (*Ghasemi et al., 2020*). These data also suggest that CTCF may play a cell-type-dependent role on *HOXA9* regulation. However, whether loss of CTCF has a direct effect on *HOXA9* expression remains to be studied. Lastly, although the clinical significance of *HOXA9* has been recognized for more than two decades, it is technically difficult to systematically discover regulators of *HOXA9* in acute leukemia owing to the lack of an endogenous reporter to dictate *HOXA9* expression.

In this work, we sought to establish an endogenous reporter system enabling real-time monitoring of *HOXA9* expression in conjunction with high-throughput CRISPR/Cas9 screening in a human B-ALL MLLr t(4;11) cell line SEM and a AML MLLr t(6;11) cell line OCI-AML2 equipped with an endogenous *HOXA9*^P2A-mCherry^ reporter allele. The *HOXA9*^P2A-mCherry^ reporter allele authentically recapitulated endogenous transcription of the *HOXA9* gene and did not affect endogenous transcription of other adjacent *HOXA* genes. To gain a global understanding of the transcription factors regulating *HOXA9* expression, we performed a CRISPR/Cas9 loss-of-function screen specifically targeting 1639 human transcription factors. Our screening robustly re-identified expected targets such as *KMT2A*, *DOT1L* and *HOXA9* itself. Surprisingly, the CRISPR screen and global depletion of CTCF via siRNA and degron-associated protein degradation all demonstrated that *HOXA9* does not downregulate upon CTCF loss. More importantly, we identified novel functional regulators of *HOXA9* including Upstream Transcription Factor 2 (USF2). USF2 depletion selectively downregulated *HOXA9* expression in MLLr leukemia cells and impaired cell growth, which could be rescued by

ectopic expression of *HOXA9* and its partner MEIS1. Thus, our *HOXA9*$^{P2A-mCherry}$ reporter lines are robust tools for discovery of novel *HOXA9* regulators.

## Results

### Establishment and characterization of the *HOXA9*$^{P2A-mCherry}$ reporter human MLLr leukemia cell line

As shown by many previous studies, *HOXA9* overexpression was observed in refractory MLL-rearranged ALL and AML patients (*Gu et al., 2019*; *Haferlach et al., 2010*; *Kohlmann et al., 2008*; *Figure 1—figure supplement 1A–C*). Therefore, we utilized our previously reported high-efficiency knock-in strategy, 'CHASE knock-in' (*Hyle et al., 2019*), to deliver the *P2A-mCherry* cassette upstream of the *HOXA9* stop codon in a patient-derived human B-ALL cell line, SEM, which has a typical B-ALL signature along with a t(4;11) translocation and maintains one single allele expression of the *HOXA* gene cluster (*Figure 1—figure supplement 1D*). Because the P2A-mediated ribosome skipping disrupts the synthesis of the glycyl-prolyl peptide bond at the C-terminus of the P2A peptide, translation leads to dissociation of the P2A peptide and its immediate downstream mCherry protein (*Kim et al., 2011*). Therefore, the knock-in allele would produce a functional HOXA9 protein under control of the endogenous promoter and intrinsic *cis*-regulatory elements while delivering a separate mCherry protein. In brief, we constructed the knock-in vector containing a *P2A-mCherry* cassette flanked with 5' and 3' *HOXA9* homology arms (HAs) of approximately 800-bps, which were cloned from SEM cells. A single guide RNA (sgRNA) and a protospacer adjacent motif (PAM) sequence targeting the genomic sequence 5' of the *HOXA9* stop codon was inserted into the 5' end of the 5' HA and 3' end of the 3' HA (*Figure 1A*). When the HA/knock-in cassette was co-electroporated with an all-in-one vector expressing wild-type Cas9 and the same *HOXA9* sgRNA, the HA/knock-in cassette was released from the donor vector with two nuclease cleavages and delivered to the target genomic region where double-strand breaks occurred. Successful knock-in cells were enriched by flow cytometry sorting for mCherry (*Figure 1B*) and characterized via genotyping PCR and Sanger sequencing (*Figure 1C*). To examine the possibility of random integration of the *P2A-mCherry* cassette, fluorescence in situ hybridization (FISH) was performed with a *P2A-mCherry* DNA probe (red) and a FITC-labeled fosmid DNA probe targeting the *HOXA9* locus (green). On-target knock-in cells displayed co-localization of red and green fluorescence without random integration signals in the rest of genome (*Figure 1D* and *Figure 1—figure supplement 2A–D*). The bulk knock-in population from SEM cells, hereafter called *HOXA9*$^{P2A-mCherry}$, was used as a reporter cell line for the entire study. Similarly, a *HOXA9*$^{P2A-mCherry}$ allele was delivered to a human MLLr AML cell line OCI-AML2 (*Figure 1—figure supplement 2E*). Many knock-in studies reported the exogenous DNA fragment may affect normal endogenous gene expression in a complex chromatin niche (*Liu et al., 2019*; *Zu et al., 2013*). Therefore, to test whether the inserted *P2A-mCherry* segment would affect the gene expression pattern of *HOXA9* and its neighboring *HOXA* cluster genes, Q-PCR analysis was conducted on both wild-type (WT) and *HOXA9*$^{P2A-mCherry}$ knock-in (KI) cells. RNA-seq data collected from SEM cells in our previous studies suggested that *HOXA7*, *HOXA9* and *HOXA10* were the only highly expressed *HOXA* genes in MLLr leukemia SEM cells (*Hyle et al., 2019*; *Figure 1E*), and that these patterns were indistinguishable between WT and KI populations, indicating the *P2A-mCherry* knock-in did not alter the gene expression landscape at the *HOXA* cluster (*Figure 1F*).

### The *HOXA9*$^{P2A-mCherry}$ reporter allele recapitulates endogenous transcription of *HOXA9* in MLLr cells

To evaluate whether the *HOXA9*$^{P2A-mCherry}$ reporter allele would faithfully respond to the transcriptional regulation of the cellular *HOXA9* promoter, we genetically perturbed or pharmaceutically inhibited *HOXA9*'s upstream regulators. Previous studies have shown that *DOT1L* and *ENL* positively regulate *HOXA9* expression in MLLr leukemia via direct occupancy on *HOXA9*'s promoter (*Zeisig et al., 2004*; *Chen et al., 2015*). Therefore, two sgRNAs targeting the coding region of *DOT1L* (sgDOT1L) and *ENL* (sgENL) were infected into the *HOXA9*$^{P2A-mCherry}$ cells expressing Cas9. Flow cytometry and Q-PCR analysis each revealed that *mCherry* and *HOXA9* expression were both downregulated by sgRNAs targeting *DOT1L* or *ENL* (*Figure 2A–D*), and that the mCherry expression correlated well with the expression of *HOXA9* (*Figure 2E*). Additionally, a DOT1L-selective

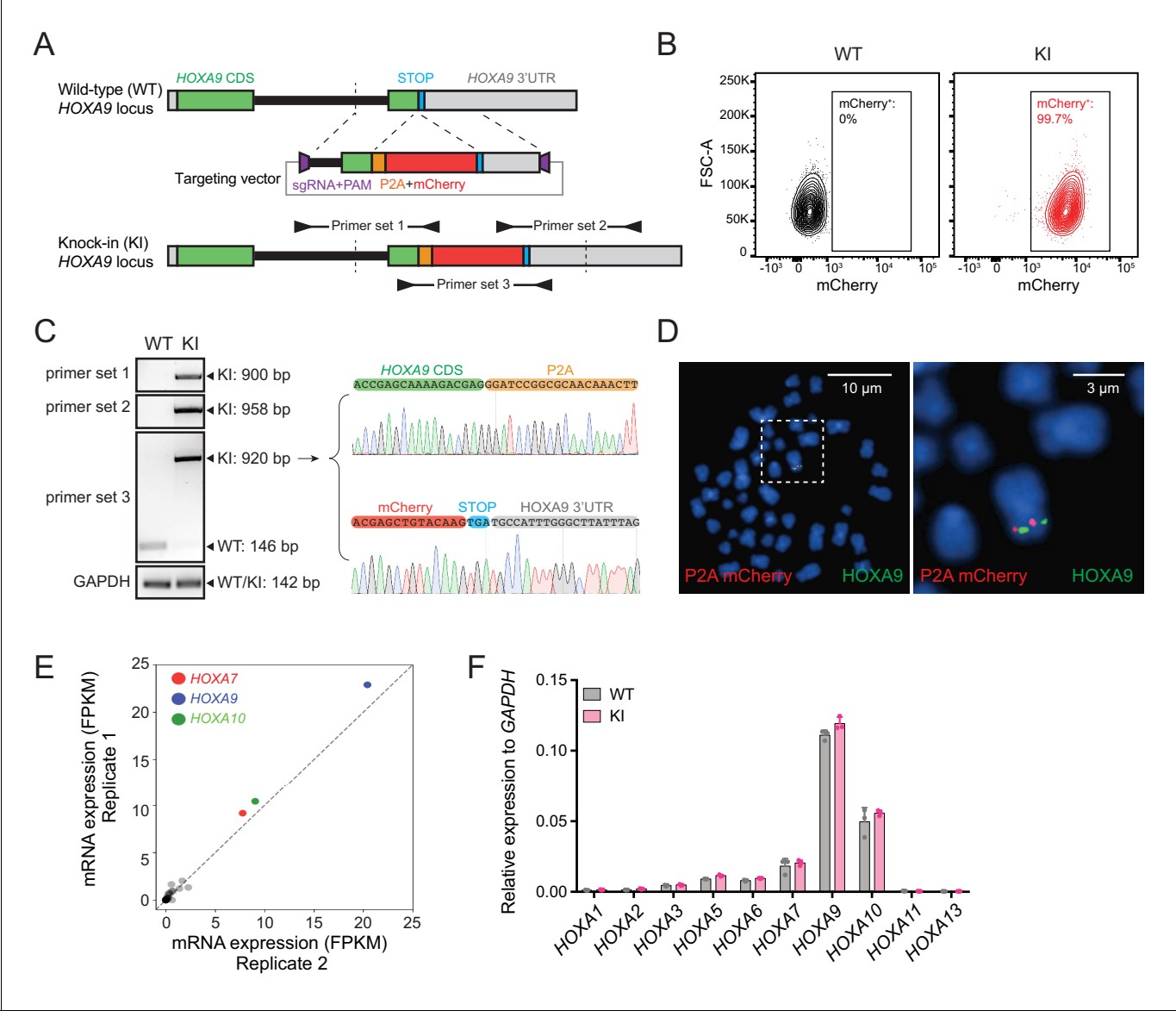

**Figure 1.** Establishment and characterization of the *HOXA9*^P2A-mCherry^ reporter human MLLr leukemia cell line. (**A**) Schematic diagram of the knock-in design and genotyping PCR primer design for the *HOXA9*^P2A-mCherry^ reporter allele. (**B**) Flow cytometry analysis of *HOXA9*^P2A-mCherry^ reporter cells. Wild-type SEM cells were used as negative controls. (**C**) Genotyping PCR products from the 5′ and 3′ knock-in boundaries were sequenced to verify the seamless knock-in of the *mCherry* reporter gene to the endogenous locus. (**D**) Fluorescence in situ hybridization of the *P2A-mCherry* knock-in cassette in *HOXA9*^P2A-mCherry^ reporter cells. The *P2A-mCherry* DNA was labeled with a red-dUTP by nick translation, and a *HOXA9 BAC* clone was labeled with a green-dUTP. The cells were then stained with 4,6-diamidino-2-phenylindole (DAPI) to visualize the nuclei. A representative metaphase cell image is shown for the pattern of hybridization (pairing of red and green signals). (**E**) RNA-seq data of all *HOXA* cluster genes were illustrated as log$_2$ (normalized numbers of FPKM) from two replicate samples of SEM cells. *HOXA7*, *HOXA9*, and *HOXA10* were highlighted by color code. (**F**) Q-PCR analysis confirmed the unaffected *HOXA* cluster gene transcription between *HOXA9*^P2A-mCherry^ reporter (KI) and WT SEM cells. Data shown are means ± SEM from replicate independent experiments. *p<0.05 of two-tailed Student's *t* test.

The online version of this article includes the following figure supplement(s) for figure 1:

**Figure supplement 1.** *HOXA9* expression profiling in leukemia.

**Figure supplement 2.** Cytogenetic characterization *HOXA9* knock-in allele in MLLr SEM and OCI-AML2 cells.

inhibitor, SGC0946 (*Yu et al., 2012*), was supplemented at different dosages for 6 days to the *HOXA9*^P2A-mCherry^ cells in culture resulting in a dosage-dependent reduction of mCherry fluorescence intensity measured by fluorescence imaging (*Figure 2F–G*) and flow cytometry (*Figure 2H*). Again, Q-PCR analysis of the DMSO- and SGC0946-treated cells showed that mRNA expression of *mCherry*

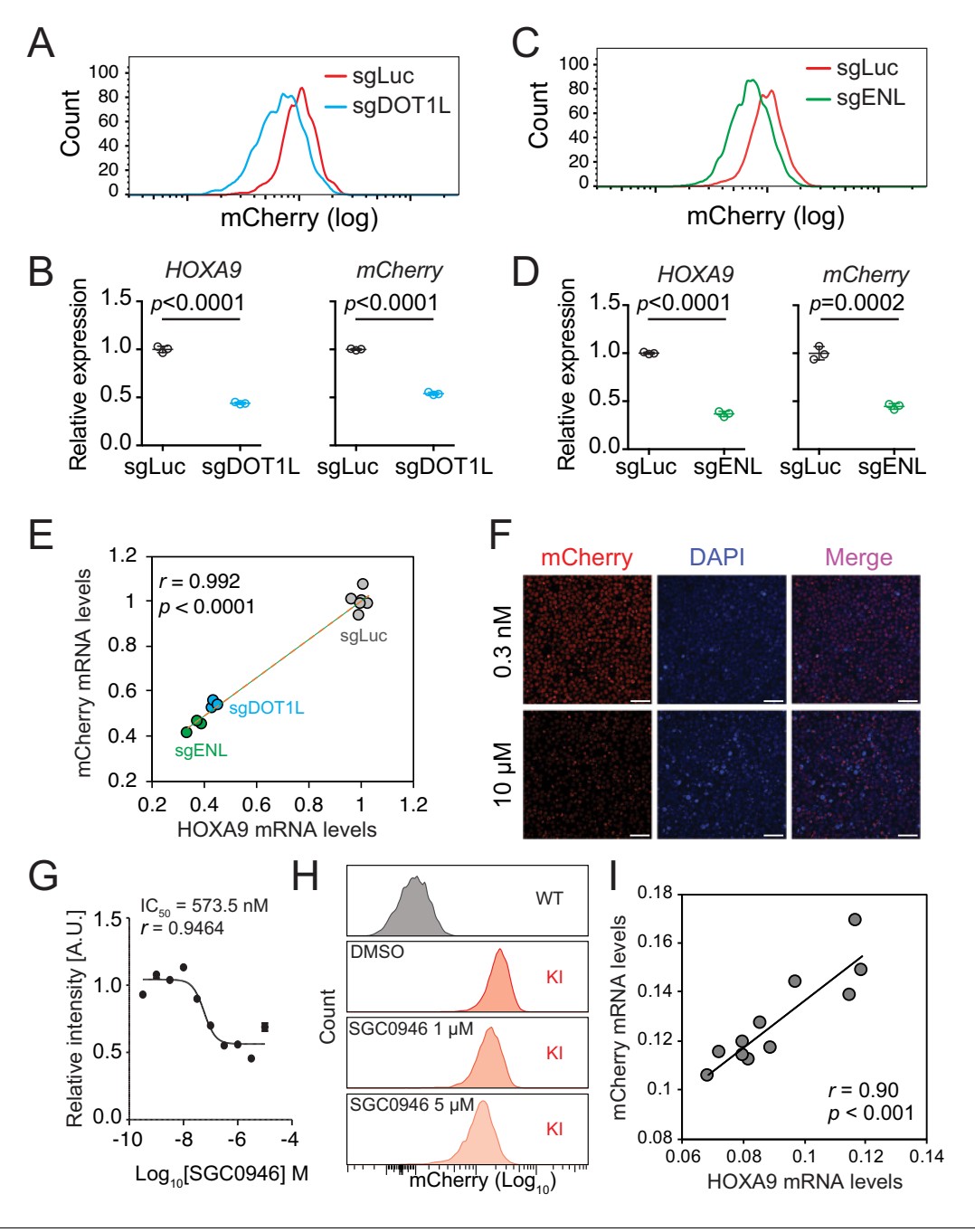

**Figure 2.** The *HOXA9*^P2A-mCherry^ reporter allele recapitulates endogenous transcription of *HOXA9* in MLLr SEM cells. (A) Flow cytometry analysis of the *HOXA9*^P2A-mCherry^ cells targeted with luciferase-sgRNA (sgLuc) and DOT1L-sgRNA (sgDOT1L). (B) Q-PCR analysis of the *HOXA9*^P2A-mCherry^ cells targeted with sgLuc and sgDOT1L by using specific primers targeting the mRNA sequences of *mCherry* and *HOXA9*. Three biological replicates were performed. Data shown are means ± SEM from replicate independent experiments. The p-value was calculated by performing a two-tailed *t*-test. (C) Flow cytometry analysis of the *HOXA9*^P2A-mCherry^ cells targeted with luciferase-sgRNA (sgLuc) and ENL-sgRNA (sgENL). (D) Q-PCR analysis of the *HOXA9*^P2A-mCherry^ cells targeted with sgLuc and sgENL by using specific primers targeting the mRNA sequence of *mCherry* and *HOXA9*. Three biological replicates were performed. The p-value was calculated by performing a two-tailed *t*-test. (E) The correlation of transcription reduction in *mCherry* and *HOXA9* in response to CRISPR–mediated targeting was calculated by Pearson's correlation test. (F) Fluorescence imaging was performed on the *HOXA9*^P2A-mCherry^ cells treated with various dosages of DOT1L inhibitor SGC0946 for six days. Representative images were shown for comparison

*Figure 2 continued on next page*

*Figure 2 continued*

between 0.3 nM and 10 µM dosages. For each dosage treatment, four replicates were conducted (scale bar 50 µm). (**G**) Fluorescence curve was generated according to mCherry intensity in response to dosage-dependent treatment of drug for 6 days. About 20,000 cells were split in each of the 384-well at the starting time point. (**H**) Flow cytometry analysis of the *HOXA9*[P2A-mCherry] cells treated with DMSO and various dosages of the DOT1L inhibitor SGC0946. (**I**) Q-PCR analysis of the *HOXA9*[P2A-mCherry] cells with or without the 6-day treatment of the DOT1L inhibitor SGC0946 by using specific primers targeting the mRNA sequences of *mCherry* and *HOXA9*. The correlation of transcription reduction in *mCherry* and *HOXA9* in response to inhibitor–mediated transcription repression was calculated by performing Pearson's correlation test.

The online version of this article includes the following figure supplement(s) for figure 2:

**Figure supplement 1.** The *HOXA9*[P2A-mCherry] reporter allele recapitulates endogenous transcription of *HOXA9* in MLLr OCI-AML2 cells.

---

was significantly correlated with that of *HOXA9* (Pearson's *r* = 0.90, p<0.001) (*Figure 2I*). Similarly, the *HOXA9*[P2A-mCherry] knock-in OCI-AML2 reporter line was also comprehensively characterized (*Figure 2—figure supplement 1A–G*). Taken together, these data confirm that the newly established *HOXA9*[P2A-mCherry] alleles were authentically controlled by the endogenous *HOXA9* promoter and its local chromatin niche.

## Pooled CRISPR/Cas9 screening identified a novel transcription factor, USF2, that regulates *HOXA9* expression

Although a few regulators of *HOXA9* in MLLr leukemia have been previously identified (*Zeisig et al., 2004*; *Collins and Hess, 2016b*; *Collins et al., 2014*; *Li et al., 2013a*; *Sun et al., 2013*; *Li et al., 2013b*; *Ogawara et al., 2015*; *de Bock et al., 2018*; *Lynch et al., 2019*), to date a comprehensive CRISPR/Cas9 screen to unbiasedly identify novel upstream regulatory factors of *HOXA9* has not been feasible owing to the lack of a reliable reporter cell line. Therefore, we combined the *HOXA9*[P2A-mCherry] reporter line and an in-house CRISPR-Cas9 sgRNA library targeting 1639 human transcription factors to identify novel regulatory effectors (*Lambert et al., 2018*). In this library, seven sgRNAs spanning multiple coding exons were designed per transcription factor, seven sgRNAs targeting DOT1L were included as a positive control, and an additional 100 non-targeting sgRNAs were included as negative controls. Two paralleled screens were performed on the same *HOXA9*[P2A-mCherry] reporter line stably expressing Cas9 and the lentiviral sgRNA library at a low M.O.I. (less than 0.3). Cells were selected with antibiotics, enriched, and fractionated by flow cytometric sorting for the top 10% (mCherry[High]) and bottom 10% (mCherry[Low]) mCherry populations, followed by genomic DNA extraction, PCR, and deep sequencing to identify differentially represented sgRNAs (*Figure 3A*). The differentially represented sgRNAs were calculated by DEseq2 analysis and combined for MAGeCK testing at the gene level (*Li et al., 2014*). The positive control genes *HOXA9* and *DOT1L* were identified among the top hits between mCherry[High] and mCherry[Low] populations, suggesting that the screening was successful (*Figure 3B*). To mitigate the possibility that key upstream regulators of HOXA9 could be missed due to a survival disadvantage, we conducted an independent CRISPR/Cas9 TF screen in *HOXA9*[P2A-mCherry] reporter SEM cells with ectopically expressed *HOXA9* together with its functional partner MEIS1. In this regard, exogenously expressed *HOXA9* could rescue the potential cell loss due to decreased *HOXA9* expression in SEM cells, while the level of endogenous *HOXA9* is still monitored by the mCherry reporter. As a result, our CRISPR screening using the *HOXA9/MEIS1* pre-rescued reporter line has identified more well-known regulators of HOXA9, which are also considered survival essential genes. Among the top 10 hits from this screen, DOT1L and HOXA9 were enriched. KMT2A, the translocation partner of MLL-AF4 in SEM cells, was identified in the HOXA9-MEIS1 rescue TF screen but not the original screen without ectopic expression of HOXA9. Notably, the MYST acetyltransferase HBO1 (also known as KAT7 or MYST2) and several members of the HBO1 protein complex, which were recently shown as critical regulators of leukemia stem cell maintenance, were also identified among the top hits (*MacPherson et al., 2020*; *Au et al., 2020*). Most importantly, USF2 was enriched among the top hits in both screens (*Figure 3B*), suggesting USF2 is likely a positive regulator with less survival essentiality compared with KMT2A. Consistent with the significant enrichment of these three candidates at the gene level, DEseq2 analysis (*Love et al., 2014*) and sgRNA enrichment plotting both

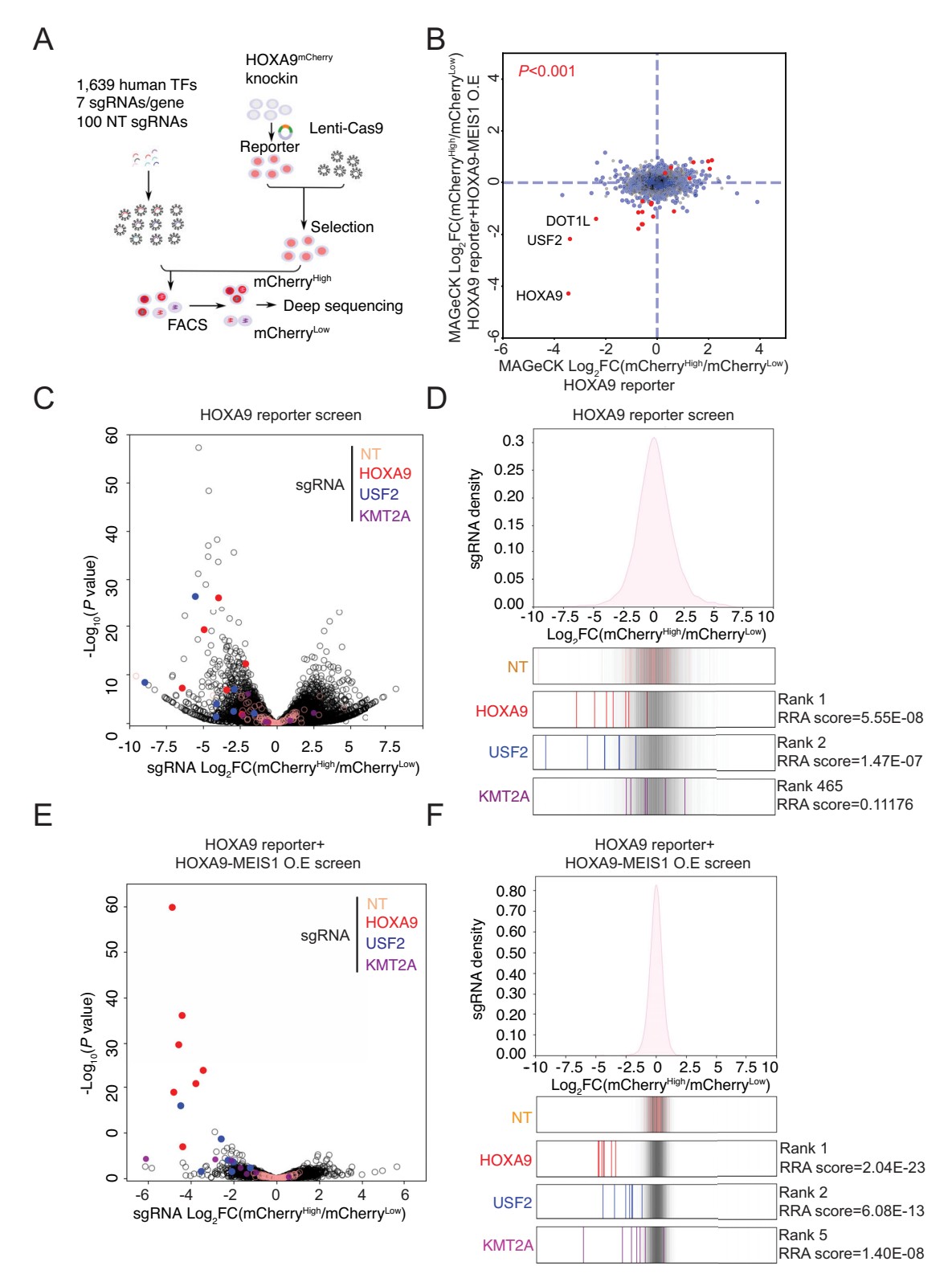

**Figure 3.** Pooled CRISPR/Cas9 screening identified a novel transcription factor, USF2, regulating *HOXA9*. (**A**) Schematic diagram of a working model of loss-of-function CRISPR screening targeting 1639 human transcription factors. (**B**) The enrichment score of seven sgRNAs against each transcription factor was combined by the MAGeCK algorithm. Positive regulators of HOXA9 were compared between parental reporter strain and HOXA9-MEIS1 overexpressed screens. Overlapped top hits including *HOXA9*, *USF2* and *DOT1L* were highlighted. (**C**) The overall distribution of all sgRNAs from the

*Figure 3 continued on next page*

*Figure 3 continued*

parental SEM HOXA9 reporter screening was shown based on the p-value and the DEseq2 score calculated by Log$_2$[Fold Change (mCherry$^{High}$/mCherry$^{Low}$)]. NT, *HOXA9*, *USF2* and *KMT2A* sgRNAs were highlighted by different color code. (D) The ratio for all sgRNAs targeting *HOXA9*, *USF2*, and *KMT2A*, are shown between mCherry$^{High}$ and mCherry$^{Low}$ sorted population. NT sgRNAs were overlaid on a gray gradient depicting the overall distribution. NT: 100 sgRNAs. Transcription factors: seven sgRNAs/each. RRA score of each gene was collected from MAGeCK analysis. (E) The overall distribution of all sgRNAs from the HOXA9-MEIS1 overexpressing SEM HOXA9 reporter screening was shown based on the p-value and the DEseq2 score calculated by Log$_2$[Fold Change (mCherry$^{High}$/mCherry$^{Low}$)]. NT, *HOXA9*, *USF2* and *KMT2A* sgRNAs were highlighted by different color code. (F) The ratio for all sgRNAs targeting *HOXA9*, *USF2*, and *KMT2A*, are shown between mCherry$^{High}$ and mCherry$^{Low}$ sorted population. NT sgRNAs were overlaid on a gray gradient depicting the overall distribution. NT: 100 sgRNAs. Transcription factors: seven sgRNAs/each. RRA score of each gene was collected from MAGeCK analysis.

The online version of this article includes the following figure supplement(s) for figure 3:

**Figure supplement 1.** CRISPR screen and data analysis by MAGeCK.
**Figure supplement 2.** CTCF is dispensable for maintaining *HOXA9* expression in MLLr SEM cells.
**Figure supplement 3.** CTCF regulates *HOXA9* expression in human colorectal cancer HCT116 cells.

suggested that most of the sgRNAs against these genes were differentially represented (*Figure 3C–F* and *Figure 3—figure supplement 1A–F*). Importantly, all the non-targeting control sgRNAs were similarly distributed across mCherry$^{High}$ and mCherry$^{Low}$ populations, indicating that the sorting-based screen did not bias the enrichment.

Interestingly, the most-characterized looping factors, CTCF and YY1, were not enriched in the *HOXA9$^{P2A-mCherry}$* reporter screen (*Figure 3B*). CTCF was reported to be essential for *HOXA9* expression by occupying the boundary sequence between *HOXA7* and *HOXA9* (CBS7/9) in MLLr AML cell line MOLM13 (*Luo et al., 2018*). CRISPR-mediated deletion of the core sequence CTCF-binding motif in CBS7/9 significantly decreased *HOXA9* expression and tumor progression (*Luo et al., 2018*; *Luo et al., 2019*). Given that CTCF is generally essential for cell survival, it is possible that cells targeted by CTCF sgRNAs in the *HOXA9$^{P2A-mCherry}$* reporter and TF screen quickly dropped out of the population and were unable to be enriched as a regulator of *HOXA9*. To mitigate the challenge, we utilized a previously described auxin-inducible degron (AID) cellular system (*Hyle et al., 2019*; *Morawska and Ulrich, 2013*; *Natsume et al., 2016*; *Nora et al., 2017*) to acutely deplete the CTCF protein in SEM cells and evaluate the immediate transcriptional response of *HOXA9* (*Figure 3—figure supplement 2A*). Upon acute depletion of CTCF via auxin (IAA) treatment in three CTCF$^{AID}$ bi-allelic knock-in clones, the protein expression of a previously identified vulnerable gene as positive control, *MYC*, was significantly inhibited (*Figure 3—figure supplement 2B*). Moreover, a Cut and Run assay using CTCF antibody for chromatin immunoprecipitation confirmed loss of CTCF occupancy throughout the *HOXA9* locus, including CBS7/9 (*Figure 3—figure supplement 2C*). However, RNA-seq data (*Figure 3—figure supplement 2D–E*) and Q-PCR analysis (*Figure 3—figure supplement 2F–G*) collected from these three clones further confirmed the observation that loss of CTCF occupancy did not correlate with a decrease in *HOXA7* or *HOXA9* expression at the mRNA level. Instead, long-term depletion of CTCF by auxin for 48 hr slightly increased the transcription of *HOXA7* and *HOXA9*. Upon washout of auxin from culture medium for an additional 48 hr, both *HOXA7* and *HOXA9* expression were restored to levels indistinguishable from those of the parental untreated cells (*Figure 3—figure supplement 2D–G*). Additionally, siRNA-mediated knock-down of CTCF in SEM cells did not change the transcription level of *HOXA7* or *HOXA9* (*Figure 3—figure supplement 2H–J*). However, suppressing CTCF in human colorectal cancer cell line HCT116 notably reduced *HOXA7* and *HOXA9* expression (*Figure 3—figure supplement 3A–E*), consistent with the finding in MLLr AML cell line MOLM13 (*Luo et al., 2018*). Collectively, these data further confirmed the results of our CRISPR screening that CTCF is not a key regulator of *HOXA9* in MLLr B-ALL SEM and likely plays a role in regulating *HOXA9* transcription in a cell-type-specific manner.

## USF2 is required to maintain *HOXA9* expression in MLLr leukemia

Aside from the positive controls confirmed from the CRISPR/Cas9 transcription factor screen in *HOXA9$^{P2A-mCherry}$* cells, the top-ranked candidate among positive regulators was USF2. To further validate the CRISPR screen result and investigate the regulatory effect of USF2 on *HOXA9* expression, we individually delivered four lentiviral sgRNAs targeting *USF2* exons 1, 2, 7, and 9 into the

*HOXA9[P2A-mCherry]* reporter line stably expressing Cas9. Similar to the results seen in sgENL targeted cells, *USF2* knock-down significantly decreased the mCherry fluorescence in a time-dependent manner compared to that of luciferase sgRNA-targeted control (sgLuc) (*Figure 4A* and *Figure 4—figure supplement 1*). Q-PCR and immunoblotting analysis further confirmed the concordant downregulation of both *HOXA9* and *mCherry* (*Figure 4B–C*). Collectively, these data suggest that USF2 positively controls *HOXA9* expression in the MLLr B-ALL SEM cell line. USF2 was reported to generally bind to a symmetrical DNA sequence (E-box motif) (5'CACGTG3') in a variety of cellular promoters (*Henrion et al., 1995*). Publicly available ChIP-seq data collected from human ES cells suggested

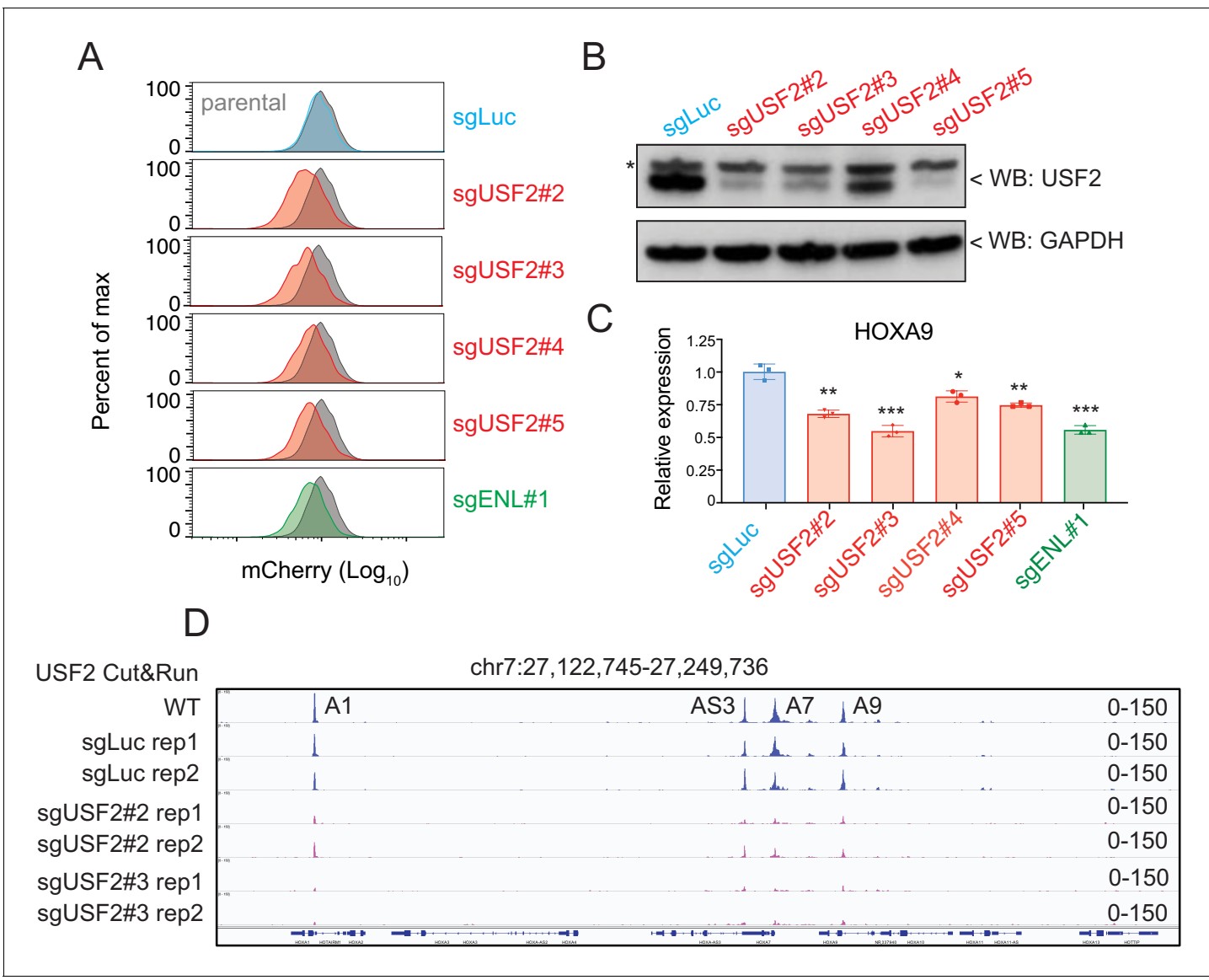

**Figure 4.** USF2 is required to maintain *HOXA9* expression in MLLr leukemia. (**A**) Flow cytometry analysis was performed at day 8 on the *HOXA9[P2A-mCherry]* cells targeted with lentiviral Cas9 and four sgRNAs against *USF2*. The sgENL-targeted cells were used as positive controls while sgLuc targeted cells were used as negative controls. (**B**) Q-PCR analysis was conducted on the USF2-targeted cells to monitor the reduction of *HOXA9*. The sgENL targeted cells were used as positive controls while sgLuc-targeted cells were used as negative controls. Data shown are means ± SEM from three independent experiments. *p<0.05, **p<0.01, ***p<0.001, two-tailed Student's *t* test. (**C**) Immunoblotting of USF2 in USF2 sgRNAs targeted cells. '*' denoted non-specific bands. (**D**) USF2 occupancy changes in sgLuc and sgUSF2-targeted SEM cells were characterized in *HOXA9* locus (A1, *HOXA1*; AS3, *HOXA-AS3*; A7, *HOXA7*; A9, *HOXA9*).

The online version of this article includes the following figure supplement(s) for figure 4:

**Figure supplement 1.** Time-course knock-down of *USF2* and consequent *HOXA9* expression analysis.

that USF2 can directly bind to the conserved E-box element at both *HOXA7* and *HOXA9* promoters (*Cheng et al., 2014*). A Cut and Run assay was performed in control sgLuc and sgUSF2 targeted SEM cells to study genome-wide USF2 occupancy. In control SEM cells, USF2 bound to *HOXA1*, *HOXA-AS3*, *HOXA7*, and *HOXA9* in *HOXA* cluster. Upon USF2 depletion, binding occupancy at these regions was significantly reduced (*Figure 4D*), further supporting the specificity of the USF2 binding identified by the Cut and Run assay. Taken together, these data suggest that USF2 could regulate *HOXA9* expression as well as other *HOXA* genes through interactions with its regulatory elements at the *HOXA* cluster gene loci.

## USF2 is an essential gene in MLLr B-ALL by controlling *HOXA9* expression

To unbiasedly evaluate the survival dependency of USF2 in SEM cells, we conducted a dropout CRISPR/Cas9 screen by targeting 1639 transcription factors. SEM cells infected with the pooled library of sgRNAs were collected at day 0 and day 12 to sequence for sgRNA distribution (*Figure 5A*). In accordance with prior genome-wide CRISPR screens and functional studies in B-ALL, many survival dependent genes were identified in the top 50 genes in our screen including *PAX5*, *DOT1L*, *ZFP64*, *YY1*, *MEF2C*, *MYC*, and *KMT2A* (*Gu et al., 2019*; *Hyle et al., 2019*; *Pridans et al., 2008*; *Lu et al., 2018*). USF2 was ranked as the top 24th essential gene in MLLr SEM cells (*Figure 5B*). Taken together, these findings suggest that the USF2/HOXA9 axis might play a role in supporting MLLr B-ALL cell proliferation. To evaluate the importance of the USF2/HOXA9 axis in MLLr B-ALL progression, we sought to investigate the knockout phenotype of USF2 in MLLr B-ALL cells. A competition-based proliferation assay was performed by infecting SEM$^{Cas9}$ cells with a lenti-viral-mCherry-sgRNAs against the *HOXA9* promoter at ~50% targeting efficiency (*Figure 5C*). The proportion of mCherry$^+$ cells were monitored over a 12-day time course (days 3, 6, 9, and 12) to investigate the proliferation disadvantage of *HOXA9* knock-down cells (*Figure 5D*). Next, the same assay was performed by infecting SEM$^{Cas9}$ cells with three individual lentiviral-mCherry-sgRNAs against USF2 (sgRNA-2,–3 and 5) at ~50% infection efficiency. As a result, the proliferation-arrested phenotype was observed in all three sgRNA targeted cells but not in cells targeted with sgLuc (*Figure 5E*). Importantly, in SEM cells constitutively expressing ectopic retroviral mouse Hoxa9 (SEM-$^{HOXA9}$), USF2 knock-down had little effect on cell growth (*Figure 5F*), suggesting that HOXA9 is a functional and essential downstream gene of USF2 in USF2-mediated leukemia propagation.

## USF1 and USF2 synergistically regulate *HOXA9* expression in MLLr leukemia

Previously, other studies identified the USF2 homolog protein USF1 shares a similar protein structure with USF2 (49, 53). USF1 and USF2 bind to the same type of E-box elements and are also able to form homo- or heterodimers (*Kumari and Usdin, 2001*; *Wang and Sul, 1995*; *Prasad and Singh, 2008*; *Spohrer et al., 2017*) suggesting that these two proteins may function in synergy to regulate *HOXA9*. Interestingly, in our HOXA9-reporter-based CRISPR screen, USF1 was also among the top 50 positive regulator genes identified (49th) (*Supplementary file 2*). To test whether USF1 and USF2 have redundant roles in regulating *HOXA9* expression, we co-delivered sgRNAs against USF2 (sgUSF2) and USF1 (sgUSF1) to the SEM *HOXA9$^{P2A-mCherry}$* reporter line stably expressing Cas9. Notably, both the flow cytometry and Q-PCR analysis confirmed a significant decrease in *HOXA9* expression with double inactivation of USF1 and USF2 compared with inactivation of USF2 alone (*Figure 5G* and *Figure 5—figure supplement 1A*), which was also supported by a synergetic effect in the competitive proliferation assay (*Figure 5H*). To further evaluate whether USF2 and USF1 could regulate HOXA9 expression in other MLLr leukemias, sgUSF2 and sgUSF1 alone or in combination were delivered into the human MLLr AML cell line OCI-AML2 which carried the MLL-AF6 transloca-tion. Similar to observations in SEM cells, USF1 or USF2 CRISPR targeting resulted in notably sup-pressed *HOXA9* expression (*Figure 5I*). In addition, USF1 and USF2 synergistically regulate *HOXA9* expression and leukemia survival in OCI-AML2 (*Figure 5J* and *Figure 5—figure supplement 1B-E*). In NOMO-1 MLLr AML cells, USF2 downregulation also notably decreased expression of *HOXA9* (*Figure 5—figure supplement 1C*) comparable to levels observed in SEM cells. Interestingly, in human MLLr AML cell line MOLM13, individual knockout of USF1 or USF2 did not affect *HOXA9* expression nor cell survival. However, USF1 and USF2 double-knockout cells demonstrated

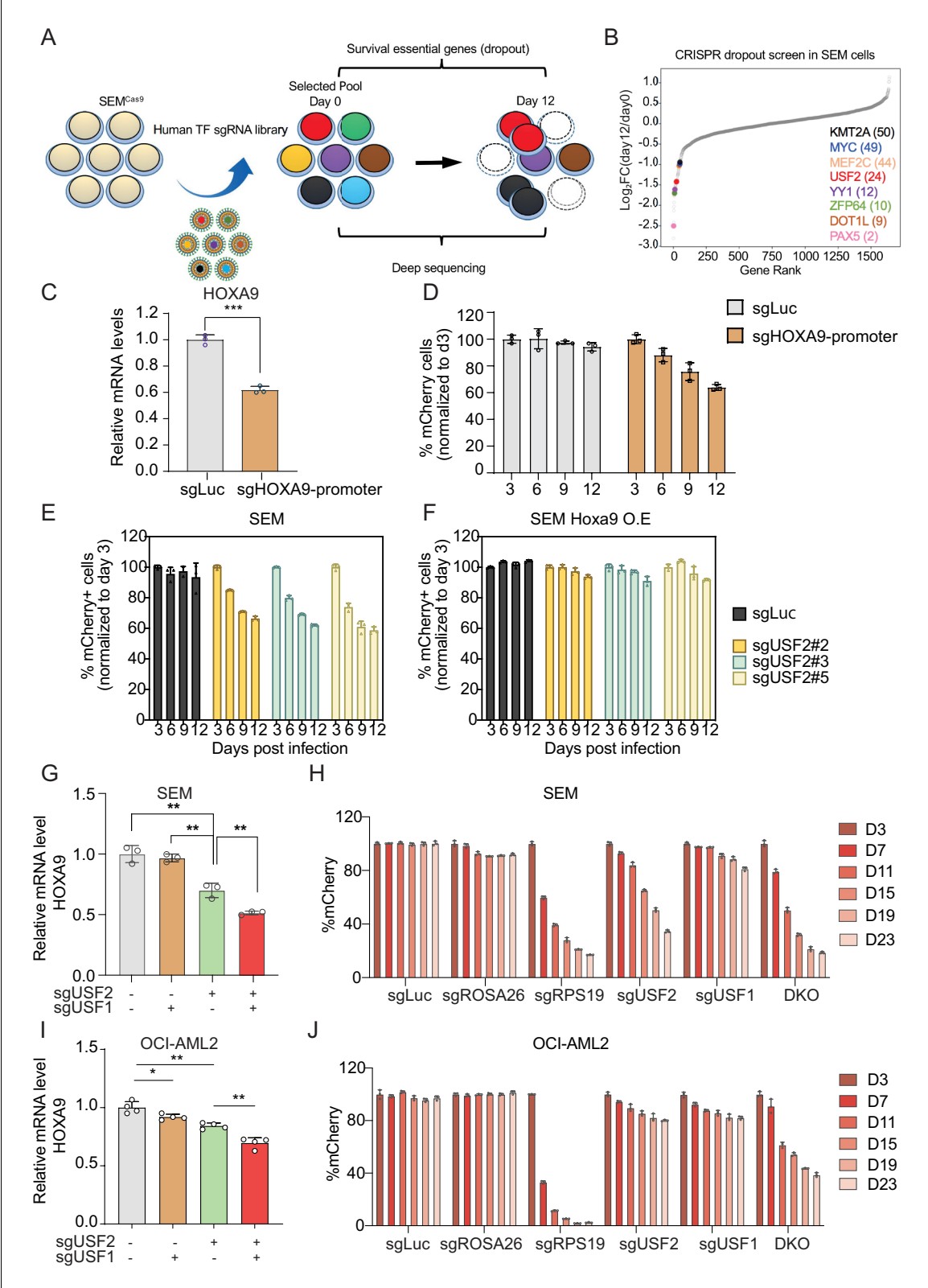

**Figure 5.** USF1 and USF2 synergistically regulate *HOXA9* expression in MLLr leukemia. (**A**) Flow diagram of dropout CRISPR screening procedure. (**B**) Gene ranking of all transcription factors from dropout screening was illustrated. The enrichment score of seven sgRNAs against each transcription factor was combined by the MAGeCK algorithm. (**C**) Q-PCR was conducted to monitor *HOXA9* expression upon CRISPR targeting on its promoter. (**D**) Competitive proliferation assay was conducted by infecting SEM[Cas9] cells with Lentiviral-mCherry-sgRNAs against *HOXA9* promoter at about 50%

*Figure 5 continued on next page*

*Figure 5 continued*

efficiency. The mCherry% was quantified every three days by flow cytometry to evaluate the growth disadvantage. (E) Competitive proliferation assay was conducted by infecting SEM$^{Cas9}$ cells with Lentiviral-mCherry-sgRNAs against luciferase (sgLuc) and USF2 (sgUSF2#2, 2#3 and 2#5) at about 50% efficiency. The mCherry% was quantified every 3 days by flow cytometry to evaluate the growth disadvantage. (F) Rescued competitive proliferation assay was conducted by infecting SEM cells overexpressing ectopic Hoxa9 with Lentiviral-mCherry-sgRNAs against luciferase (sgLuc) and USF2 (sgUSF2#2, 2#3 and 2#5) at about 50% efficiency. The mCherry% was quantified every 3 days by flow cytometry to evaluate the growth disadvantage. (G) Q-PCR analysis was conducted on the sgUSF2, sgUSF1 and sgUSF1/2-targeted SEM cells to monitor the reduction of *HOXA9*. Data shown are means ± SEM from three independent experiments. **p<0.01, two-tailed Student's *t* test. (H) Competitive proliferation assay was conducted by infecting SEM$^{Cas9}$ cells with Lentiviral-mCherry-sgLuc, sgUSF1, sgUSF2, and sgUSF1/2 (DKO) at about 50% efficiency. The mCherry% was quantified at days 3, 7, 11, 15, 19, and 23 by flow cytometry to evaluate the growth disadvantage. A guide RNA targeting the survival essential gene RPS19 was included as a positive control for Cas9 activity. Guide RNAs targeting Luciferase gene (sgLuc) and the human ROSA26 gene (sgROSA26) were included as a negative control. (I) Q-PCR analysis was conducted on the sgUSF2, sgUSF1 and sgUSF1/2 targeted OCI-AML2 cells to monitor the reduction of *HOXA9*. Data shown are means ± SEM from three independent experiments. *p<0.05, **p<0.01, two-tailed Student's *t* test. (J) Competitive proliferation assay was conducted by infecting OCI-AML2$^{Cas9}$ cells with Lentiviral-mCherry-sgLuc, sgUSF1, sgUSF2, and sgUSF1/2 (DKO) at about 50% efficiency. The mCherry% was quantified at days 3, 7, 11, 15, 19, and 23 by flow cytometry to evaluate the growth disadvantage. A guide RNA targeting the survival essential gene RPS19 was included as a positive control for Cas9 activity. Guide RNAs targeting Luciferase gene (sgLuc) and the human ROSA26 gene (sgROSA26) were included as negative controls.

The online version of this article includes the following figure supplement(s) for figure 5:

**Figure supplement 1.** USF2 depletion in MLLr leukemia cells.
**Figure supplement 2.** USF2 depletion in non-MLLr leukemia cells.
**Figure supplement 3.** Transcriptional correlation between *USF2* and *HOXA9* in patient cohorts.

suppressed *HOXA9* expression and reduced survival (*Figure 5—figure supplement 1F-G*). Taken together, the data suggests that loss of one *USF* family member gene may lead to varying degrees of compensatory regulation of *HOXA9* by the untargeted *USF* gene, whereas loss of both *USF* genes results in a more robust abrogation of *HOXA9* expression.

To examine if USF2 regulation of *HOXA9* expression was unique to MLLr leukemias, we used two sgRNAs, sgUSF2#2 and sgUSF2#3, to knockdown USF2 expression in two human non-MLLr leukemia cell lines, OCI-AML3 and U937, which both express *HOXA9*. Upon complete USF2 depletion, *HOXA9* expression remained unchanged, suggesting the USF2/HOXA9 axis may function in a MLLr-dependent manner (*Figure 5—figure supplement 2A-D*). Lastly, a transcriptome analysis from the to-date largest human B-ALL transcriptome cohort (N = 1988 patients) (*Gu et al., 2019*) identified *USF2* expression to be significantly correlated with *HOXA9* in MLLr-subtype patients (N = 136 patients) (*Figure 5—figure supplement 3A-D*) highlighting that the USF2 and *HOXA9* regulation axis could have clinical relevance for patients in this specific subtype.

## Discussion

*HOX* genes are a cluster of genes strictly regulated in development by various transcription and epigenetic modulators. Dysregulation of *HOX* genes has been frequently linked to human diseases, particularly cancer. Here, we focus on *HOXA9*, the aberrant expression of which is one of the most significant features in the most aggressive human leukemias. The *HOXA9$^{P2A-mCherry}$* knock-in MLLr cell line derived in this study fully recapitulated transcriptional regulation of the endogenous gene. Previously, Godmin, et al. derived two mouse strains by delivering the in-frame GFP cassette to two different murine *Hox* genes, *Hoxa1* and *Hoxc13*, to visualize the proteins during mouse embryogenesis (*Godwin et al., 1998*). Although this previous study certainly added to the repertoire of research tools available to investigate *HOXA*-related gene expression and gene function, our *HOXA9* reporter cell line provides a unique intrinsic cellular model with which to study transcriptional regulation of human *HOXA9* directly. Additionally, the CHASE-knock-in protocol developed to generate the *HOXA9* reporter is user-friendly, highly efficient, robust to reproduce and could be easily adapted to a wide variety of HOXA9-driven human leukemia cell models and other HOXA9-expressing cancer types.

In mammalian cells, each chromosome is hierarchically organized into hundreds of megabase-sized TADs (*ENCODE Project Consortium, 2012*; *Ji et al., 2016*; *Rowley et al., 2017*; *Rowley and Corces, 2018*), each of which is insulated by the boundary elements. Within the TAD scaffold, promoter/enhancer physical contacts intricately regulate gene expression (*Pombo and Dillon, 2015*).

Intra-TAD chromatin interactions can be facilitated by a pair of CTCF-binding sites engaged in contact with each other when they are in a convergent linear orientation (*Rao et al., 2014*; *Vietri Rudan et al., 2015*). The *HOXA9* cluster is located on the TAD boundary, providing an opportunity to interact with neighboring genomic elements. However, because of the low resolution of publicly available Hi-C data and the lack of DpnI restriction enzyme sites within the *HOXA* gene cluster that are necessary to generate high-quality 3C libraries, the impact of chromatin interaction regulation of *HOXA9* remains unclear. Using a chromosome conformation capture-based PCR assay and CRISPR-mediated deletion of a minimal CTCF-binding motif between *HOXA7* and *HOXA9* (CBS7/9), Luo and colleagues proposed that the CTCF boundary was crucial for higher order chromatin organization by showing the depletion of CBS7/9 disrupted chromatin interactions and significantly reduced *HOXA9* transcription in MLLr AML MOLM13 cells with t(9;11) (*Luo et al., 2018*; *Luo et al., 2019*). In our study, the loss-of-function results from auxin-inducible degradation of CTCF, siRNA-mediated CTCF knock-down, and the unbiased transcription factor screening suggested that CTCF is not required to maintain *HOXA9* expression in SEM cells with MLLr with t(4;11). We speculate that the discrepancy could be due to the following reasons. Although both cell lines carried the MLLr translocation as a driver oncogenic mutation, MOLM13 and SEM were classified as AML and B-ALL, respectively. Besides the lineage difference, SEM cells are also less sensitive to many well-known pharmaceutical inhibitors including JQ1 and DOT1L inhibitor. Therefore, we hypothesized that other as yet to be identified looping factors might be involved in the transcriptional regulation of the *HOXA9* locus in MLLr SEM cells, and that CTCF regulates HOXA9 expression in a cell-type-specific context.

By performing unbiased CRISPR screens designed to target 1639 known human transcription factors in a *HOXA9*[P2A-mCherry] reporter cell line, we identified USF2 as a novel regulator of *HOXA9*. In addition, two known *HOXA9* regulators, *HOXA9* and *DOT1L*, were identified among the top hits supporting the reliable sensitivity of both the reporter system and the CRISPR screening strategy. USF2 is a ubiquitously expressed basic helix-loop-helix-leucine-zip transcription factor that generally recognizes E-box DNA motifs (*Henrion et al., 1995*; *Groenen et al., 1996*; *Luo and Sawadogo, 1996*). USF1 and USF2 usually form homo- or heterodimers to modulate gene expression (*Kumari and Usdin, 2001*). Interestingly, USF1 was also enriched in our CRISPR screening. Moreover, the function of USF2 in controlling leukemia progression has not been reported. Although our study identified the regulatory function of USF1/USF2 on *HOXA9* maintenance and leukemia cell survival in MLLr B-ALL and AML cell lines, other HOXA9-independent functions of USF1/2 cannot be excluded and requires further studies.

In summary, we revealed that candidate transcription factors identified from the CRISPR/Cas9 screen including USF2 and USF1, regulate *HOXA9* thereby providing a more comprehensive understanding about how the *HOXA9* locus is regulated in human cancer cells. Given the well-recognized role of *HOXA9* in hematopoietic malignancies, we anticipate the *HOXA9* reporter cells will advance many lines of investigation including drug screening and the identification of concordant epigenetic modifiers/transcription factors that are required for activation and maintenance of *HOXA9* expression in leukemia progression. Collectively, these efforts would clarify the molecular mechanisms underlying aberrant *HOXA9* activation in leukemias, thus providing the foundation to develop clinically relevant therapies to target the expression and/or function of *HOXA9* in leukemia patients.

# Materials and methods

## Key resources table

| Reagent type (species) or resource | Designation | Source or reference | Identifiers | Additional information |
|---|---|---|---|---|
| Cell line (*Homo sapiens*) | SEM | DSMZ | ACC546 | CVCL_0095 |
| Cell line (*Homo sapiens*) | NOMO-1 | DSMZ | ACC542 | CVCL_1609 |
| Cell line (*Homo sapiens*) | OCI-AML2 | DSMZ | ACC99 | CVCL_1619 |

*Continued on next page*

*Continued*

| Reagent type (species) or resource | Designation | Source or reference | Identifiers | Additional information |
|---|---|---|---|---|
| Cell line (*Homo sapiens*) | OCI-AML3 | DSMZ | ACC582 | CVCL_1844 |
| Cell line (*Homo sapiens*) | MOLM13 | DSMZ | ACC554 | CVCL_2119 |
| Cell line (*Homo sapiens*) | U937 | ATCC | CRL-1593.2 | CVCL_0007 |
| Cell line (*Homo sapiens*) | 293T | ATCC | CRL-3216 | CVCL_0063 |
| Cell line (*Homo sapiens*) | SEM-HOXA9$^{P2A-mCherry}$ | This eLife study | Reporter derived from SEM cells via knock-in | Cell line is available upon request to Dr. Chunliang Li |
| Cell line (*Homo sapiens*) | OCI-AML2-HOXA9$^{P2A-mCherry}$ | This eLife study | Reporter derived from OCI-AML2 cells via knock-in | Cell line is available upon request to Dr. Chunliang Li |
| Antibody | Anti-USF2 (Rabbit polyclonal) | Novus | NBP1-92649 | IP, IB (1:2,000) AB_11007053 |
| Antibody | Anti-USF1 | Proteintech | 22327–1-AP | IB (1: 2,000) AB_2060867 |
| Antibody | Anti-CTCF (Rabbit polyclonal) | Abcam | ab70303 | IB (1:1,000) AB_1209546 |
| Antibody | Anti-MYC (Rabbit polyclonal) | Cell Signaling Technology | 9402 | IB (1:1000) AB_2151827 |
| Antibody | Anti-GAPDH | Thermo Fisher Scientific | AM4300 | IB (1:10,000) AB_437392 |
| Antibody | Anti-Vinculin | Proteintech | 26520–1-AP | IB (1:2,000) AB_2868558 |
| Sequence-based reagent | U6-Forward sequencing primer | This paper | sgRNA sequencing primer | 5'GAGGGCCTATTTCCCATGAT3' |
| Sequence-based reagent | sgRNA sequence | This paper | sgRNA targeting HOXA9 on C-terminus | 5'AAAGACGAGTGATGCCATTT3' |
| Sequence-based reagent | HOXA9 5'HA cloning primer F | This paper | HOXA9 knockin reporter cloning | 5'GGCCGATTCCTTCCACTTCT3' |
| Sequence-based reagent | HOXA9 5'HA cloning primer R | This paper | HOXA9 knockin reporter cloning | 5'TCACTCGTCTTTTGCTCGGT3' |
| Sequence-based reagent | HOXA9 3'HA cloning primer F | This paper | HOXA9 knockin reporter cloning | 5'ACCGAGCAAAAGACGAGTGA3' |
| Sequence-based reagent | HOXA9 3'HA cloning primer R | This paper | HOXA9 knockin reporter cloning | 5'CACTGTTCGTCTGGTGCAAA3'. |
| Sequence-based reagent | Infusion cloning F | This paper | HOXA9 knockin reporter cloning | 5'AAGACCGAGCAAAAGACGAGGGATCCGGCGCAACAAACTT3' |
| Sequence-based reagent | Infusion cloning R | This paper | HOXA9 knockin reporter cloning | 5'AATAAGCCCAAATGGCATCACTTGTACAGCTCGTCCATGC3' |
| Sequence-based reagent | Infusion cloning of mCherry F | This paper | HOXA9 knockin reporter cloning | 5'AAAGACGAGTGATGCCATTTGGGATGAGGCTGCGGGCGAC3' |

*Continued on next page*

*Continued*

| Reagent type (species) or resource | Designation | Source or reference | Identifiers | Additional information |
|---|---|---|---|---|
| Sequence-based reagent | Infusion cloning of mCherry R | This paper | HOXA9 knockin reporter cloning | 5′AAAGACGAGTGA TGCCATTTGGGTATA TATACAATAGACA AGACAGGAC3′ |
| Sequence-based reagent | DOT1L-sgRNA | This paper | sgRNA sequence | 5′TCAGCTTCGAG AGCATGCAG3′ |
| Sequence-based reagent | ENL-sgRNA | This paper | sgRNA sequence | 5′TCACCTGGAC GGTGCACTGG3′ |
| Sequence-based reagent | USF2-sgRNA#2 | This paper | sgRNA sequence | 5′AGAAGAGCCC AGCACAACGA3′ |
| Sequence-based reagent | USF2-sgRNA#3 | This paper | sgRNA sequence | 5′TGTTTTCCGC AGTGGAGCGG3′ |
| Sequence-based reagent | USF2-sgRNA#4 | This paper | sgRNA sequence | 5′CCGGGGATC TTACCTGGCGG3′ |
| Sequence-based reagent | USF2-sgRNA#5 | This paper | sgRNA sequence | 5′CAGCCACGAC AAGGGACCCG3′ |
| Sequence-based reagent | USF1-sgRNA | This paper | sgRNA sequence | 5′CTATACTTAC TTCCCCAGCA3′ |
| Sequence-based reagent | Luciferase-sgRNA | This paper | sgRNA sequence | 5′CCCGGCGCCA TTCTATCCGC3′ |
| Sequence-based reagent | ROSA26-sgRNA | This paper | sgRNA sequence | 5′ACCTACCAC ACTAGCCCGA3′ |
| Sequence-based reagent | RPS19-sgRNA | This paper | sgRNA sequence | 5′GTAGAACCAG TTCTCATCGT3′ |
| Sequence-based reagent | HOXA9-promoter sgRNA | This paper | sgRNA sequence | 5′GATTTCATGT AACAACTTGG3′ |
| Sequence-based reagent | CTCF-F | This study | Q-PCR primer | 5′TTTGTCTGTTC TAAGTGTGGGAAA3′ |
| Sequence-based reagent | CTCF-R | This study | Q-PCR primer | 5′TTAGAGCGCAT CTTTCTTTTTCTT3′ |
| Sequence-based reagent | GAPDH-F | This study | Q-PCR primer | 5′AGGGCTGCTTT TAACTCTGGT3′ |
| Sequence-based reagent | GAPDH-R | This study | Q-PCR primer | 5′CCCCACTTGATT TTGGAGGGA3′ |
| Sequence-based reagent | ACTB-F | This study | Q-PCR primer | GAGCACAGAGC CTCGCCTTT |
| Sequence-based reagent | ACTB-R | This study | Q-PCR primer | GAGCGCGGCG ATATCATCA |
| Sequence-based reagent | HOXA1-F | This study | Q-PCR primer | 5′CCAGCCACCAA GAAGCCTGT3′ |
| Sequence-based reagent | HOXA1-R | This study | Q-PCR primer | 5′CCAGTTCCGT GAGCTGCTTG3′ |
| Sequence-based reagent | HOXA2-F | This study | Q-PCR primer | 5′ACAGCGAAGGGA AATGTAAAAGC3′ |
| Sequence-based reagent | HOXA2-R | This study | Q-PCR primer | 5′GGGCCCCAG AGACGCTAA3′ |
| Sequence-based reagent | HOXA3-F | This study | Q-PCR primer | 5′TGCAAAAGCG ACCTACTACGA3′ |
| Sequence-based reagent | HOXA3-R | This study | Q-PCR primer | 5′CGTCGGCG CCCAAAG3′ |
| Sequence-based reagent | HOXA4-F | This study | Q-PCR primer | 5′CGTGGTGTAC CCCTGGATGA3′ |

*Continued on next page*

*Continued*

| Reagent type (species) or resource | Designation | Source or reference | Identifiers | Additional information |
|---|---|---|---|---|
| Sequence-based reagent | HOXA4-R | This study | Q-PCR primer | 5'AAGACCTGCT GCCGGGTGTA3' |
| Sequence-based reagent | HOXA5-F | This study | Q-PCR primer | 5'TCTACCCCTG GATGCGCAAG3' |
| Sequence-based reagent | HOXA5-R | This study | Q-PCR primer | 5'AATCCTCCTTC TGCGGGTCA3' |
| Sequence-based reagent | HOXA6-F | This study | Q-PCR primer | 5'TGGATGCAGC GGATGAACTC3' |
| Sequence-based reagent | HOXA6-R | This study | Q-PCR primer | 5'CCGTGTCAGGT AGCGGTTGA3' |
| Sequence-based reagent | HOXA7-F | This study | Q-PCR primer | 5'TCTGCAGTGAC CTCGCCAAA3' |
| Sequence-based reagent | HOXA7-R | This study | Q-PCR primer | 5'AGCGTCTGGT AGCGCGTGTA3' |
| Sequence-based reagent | HOXA9-F | This study | Q-PCR primer | 5'AAAAACAACC CAGCGAAGGC3' |
| Sequence-based reagent | HOXA9-R | This study | Q-PCR primer | 5'ACCGCTTTTT CCGAGTGGAG3' |
| Sequence-based reagent | HOXA10-F | This study | Q-PCR primer | 5'CCTTCCGAGAG CAGCAAAGC3' |
| Sequence-based reagent | HOXA10-R | This study | Q-PCR primer | 5'CAGCGCTTCT TCCGACCACT3' |
| Sequence-based reagent | HOXA11-F | This study | Q-PCR primer | 5'ACAGGCTTTCG ACCAGTTTTTC3' |
| Sequence-based reagent | HOXA11-R | This study | Q-PCR primer | 5'CCTTCTCGGC GCTCTTGTC3' |
| Sequence-based reagent | HOXA13-F | This study | Q-PCR primer | 5'ACTCTGCCCGA CGTGGTCTC3' |
| Sequence-based reagent | HOXA13-R | This study | Q-PCR primer | 5'TTCGTGGCGT ATTCCCGTTC3' |
| Sequence-based reagent | mCherry-F | This study | Q-PCR primer | 5'CACTACGACG CTGAGGTCAA3' |
| Sequence-based reagent | mCherry-R | This study | Q-PCR primer | 5'TAGTCCTCGTT GTGGGAGGT3' |
| Sequence-based reagent | siRNA: nontargeting control | Thermo Fisher | siRNA oligo | Silencer Select |
| Sequence-based reagent | siRNA: CTCF | Thermo Fisher | siRNA oligo | Silencer Select |
| Sequence-based reagent | DNA oligo pool | CustomArray | | sgRNA synthesis |
| Commercial assay or kit | In-Fusion HD Cloning | Clontech | 638909 | |
| Commercial assay or kit | Polybrene | EMD Millipore | TR-1003-G | |
| Commercial assay or kit | DAPI | Sigma | D9542-10MG | |
| Commercial assay or kit | Lonza nucleofector Kit | Lonza | VCA-1003 | |
| Commercial assay or kit | Q5 High-Fidelity DNA Polymerase | New England Biolabs | M0491L | |

*Continued*

| Reagent type (species) or resource | Designation | Source or reference | Identifiers | Additional information |
|---|---|---|---|---|
| Commercial assay or kit | CloneAMP HiFI PCR Premix | Clontech | 639298 | |
| Commercial assay or kit | Quick-DNA Miniprep Kit | Zymo | D3025 | |
| Commercial assay or kit | NEB Next UltraII DNA Library Prep Kit | NEB | E7645S | |
| Commercial assay or kit | ZymoPURE II Plasmid Midiprep Kit | Zymo | D4201 | |
| Commercial assay or kit | TRIzol | Thermo Fisher Scientific | 15596026 | |
| Commercial assay or kit | High-Capacity cDNA Reverse Transcription Kit | Applied Biosystems | 4374966 | |
| Commercial assay or kit | FAST SYBR Green Master Mix | Applied Biosystems | 4385612 | |
| Recombinant DNA reagent | pSpCas9 (BB)—2A-GFP | Addgene | 48138 | PX458 |
| Recombinant DNA reagent | TOPO-cloning vector | Thermo Fisher Scientific | 450641 | |
| Recombinant DNA reagent | Lenti-Cas9-Blast plasmid | Addgene | 83480 | |
| Recombinant DNA reagent | Lenti-Guide-Puro plasmid | Addgene | 52963 | |
| Recombinant DNA reagent | LRCherry2.1 | Addgene | 108099 | |
| Recombinant DNA reagent | LRNeo-2.1 vector | This study | | Subclone from LRCherry2.1 |
| Recombinant DNA reagent | Lenti-Guide-Puro-IRES-CFP plasmid | This study | | Subclone from Lenti-Guide-Puro |
| Recombinant DNA reagent | HOXA9-MEIS1 OE | This study | | Subclone from mouse cDNA |
| Recombinant DNA reagent | mHoxa9 OE | This study | | Subclone from mouse cDNA |
| Chemical compound, drug | Puromycin | InvivoGen | ant-pr-1 | |
| Chemical compound, drug | Neomycin | GeminiBio | 400–121P | |
| Chemical compound, drug | Blasticidine | Gibco | A1113903 | |
| Chemical compound, drug | SGC0946 | MedChemExpress | HY-15650 | DOT1L inhibitor |
| Software, algorithm | Fluorescene Imaging | Perkin Elmer | | Columbus Image Data Storage and Analysis system |
| Software, algorithm | MAGeCK | https://sourceforge.net/p/mageck/wiki/Home/ | PMID:25476604 | |

*Continued on next page*

*Continued*

| Reagent type (species) or resource | Designation | Source or reference | Identifiers | Additional information |
|---|---|---|---|---|
| Software, algorithm | MACS2 | https://github.com/macs3-project/MACS; *Zhang et al., 2008* | | |
| Software, algorithm | Cutadapt | https://cutadapt.readthedocs.io/en/v1.9.1/installation.html | 1.9.1 | |
| Software, algorithm | BWA | https://github.com/lh3/bwa/releases; *Li, 2013* | 0.7.17-r1188 | |
| Software, algorithm | Samtools | http://www.htslib.org/ | Htslib 1.6 | |
| Software, algorithm | IGV | http://software.broadinstitute.org/software/igv/ | IGV2.3.97 | |
| Software, algorithm | ChIPseeker | https://guangchuangyu.github.io/software/ChIPseeker/; *Yu et al., 2015* | | |
| Software, algorithm | TRANSFAC | http://gene-regulation.com/pub/databases.html | | |
| Software, algorithm | JASPAR | http://jaspar.genereg.net/ | 8th release (**2020**) | |
| Software, algorithm | FIMO | http://meme-suite.org/doc/fimo.html | | |
| Software, algorithm | DESeq2 | https://bioconductor.org/packages/release/bioc/html/DESeq2.html | | |
| Software, algorithm | GraphPad Prism | version 8.0 | | |
| Software, algorithm | Flowjo | version 10.0 | | |
| Software, algorithm | Bowtie | http://bowtie-bio.sourceforge.net/index.shtml | | |
| Software, algorithm | BamCoverage | https://deeptools.readthedocs.io/en/develop/content/tools/bamCoverage.html | | |

## Cell culture

SEM cells (ACC-546, DSMZ), OCI-AML2 (ACC-99, DSMZ), Cas9-expressing OCI-AML3 (originally from ACC-582, a kind gift from Dr. Christopher Vakoc) and MOLM13 (ACC-554, DSMZ) were maintained in RPMI-1640 medium (Lonza) containing 10% fetal bovine serum (FBS) (HyClone), and 1% penicillin/streptomycin (Thermo Fisher Scientific) at 37˚C, 5% $CO_2$ atmosphere and 95% humidity. Basal medium for culturing 293 T cells is DMEM (HyClone). All passages of cells used in this study were mycoplasma-free. Cell identity was confirmed by STR analysis.

## Vector construction

A pair of oligomers containing a 20 bp sgRNA (5'-AAAGACGAGTGATGCCATTT-3') sequence targeting the surrounding genomic segment of *HOXA9* stop codon was synthesized (Thermo Fisher Scientific) and cloned into the all-in-one vector, pSpCas9(BB)−2A-GFP (Addgene #48138) between *BsmBI* sites. Correct clones were screened and confirmed by Sanger sequencing with the U6-Forward sequencing primer (5'-GAGGGCCTATTTCCCATGAT-3'). To construct a CHASE-knock-in donor vector delivering a *P2A-mCherry* DNA segment to the endogenous *HOXA9* locus, a two-step cloning protocol was used. The ~800 bp 5' and 3' homology arm (HA) flanking the endogenous

sgRNA target was amplified from SEM cells. The 5′ HA PCR primer sequences are 5′-GGCCGA TTCCTTCCACTTCT-3′ and 5′-TCACTCGTCTTTTGCTCGGT-3′, and the 3′ HA PCR primer sequences are 5′-ACCGAGCAAAAGACGAGTGA-3′ and 5′-CACTGTTCGTCTGGTGCAAA-3′. The *P2A-mCherry* DNA fragment was amplified from p16$^{INK4A}$-P2A-mCherry knock-in donor vector (*Zhang et al., 2019*) using a pair of primers containing overlapping sequences of 5′ HA or 3′ HA for in-fusion cloning (forward primer: 5′-AAGACCGAGCAAAAGACGAGGGATCCGGCGCAACAAAC TT-3′; reverse primer: 5′- AATAAGCCCAAATGGCATCACTTGTACAGCTCGTCCATGC-3′). The 5′ HA-P2A-mCherry-3′ HA in-fusion cloning product was further supplemented with 23 bp target sgRNA and PAM sequences at both 5′ and 3′ ends through PCR amplification using primers 5′-AAA-GACGAGTGATGCCATTTGGGATGAGGCTGCGGGCGAC-3′ and 5′-AAAGACGAGTGATGCCA TTTGGGTATATATACAATAGACAAGACAGGAC-3′. The cloning PCR reactions were performed using Q5 High-Fidelity DNA Polymerase (New England Biolabs # M0491L), and the cycling parameters were as follows for all cloning: 98℃ for 30 s, followed by 98℃ for 15 s, 72℃ for 20 s, and 72℃ for 30 s per kb for 40 cycles. The final PCR product was conducted into TOPO cloning vector (Thermo Fisher Scientific #450641). Sanger sequencing was performed to ensure that the knock-in DNA was cloned in-frame with the HAs. The Lenti-Cas9-Blast plasmid (Addgene #83480) and the Lenti-Guide-Puro plasmid (Addgene #52963) were purchased from Addgene. For candidate validation of CRISPR screen, sgRNA sequences against *DOT1L* (5′-TCAGCTTCGAGAGCATGCAG-3′), *ENL* (5′-TCACCTGGACGGTGCACTGG-3′), *USF2* (#2: 5′-AGAAGAGCCCAGCACAACGA-3′, #3: 5′-TG TTTTCCGCAGTGGAGCGG-3′, #4: 5′-CCGGGGATCTTACCTGGCGG-3′, and #5: 5′-CAGCCACGA-CAAGGGACCCG-3′) were cloned into an in-house-made Lenti-Guide-Puro-IRES-CFP vector. The sgRNA sequence against *USF1* (3#, 5′-CTATACTTACTTCCCCAGCA-3′) was cloned into an in-house-made LRNeo-2.1 vector in which the mCherry-expressing cassette of LRCherry2.1 (Addgene #108099) was replaced by Neomycin. For competitive proliferation assay, sgRNAs against Luciferase (Luc)(5′-CCCGGCGCCATTCTATCCGC-3′) and USF2 (#2, #3 and #5 as above) were cloned into mCherry-expressing LRCherry2.1 (Addgene #108099) vector.

## Generation of a *HOXA9$^{P2A-mCherry}$* reporter allele

SEM and OCI-AML2 were electroporated by using the Nucleofector-2b device (Lonza) with the V-kit and program X-001. For *HOXA9$^{P2A-mCherry}$* knock-in delivery, 2.5 μg of the donor plasmid and 2.5 μg of the CRISPR/Cas9-HOXA9-C-terminus-sgRNA all-in-one plasmid were used for 5 million SEM cells. Twenty-four hours after transfection, cells were sorted for the GFP fluorescent marker linked to Cas9 expression vector to enrich the transfected cell population. After the sorted cells recovered in culture for up to 3 weeks, a second sort was performed to select cells for successful knock-in by sorting for cells expressing the knock-in mCherry fluorescent marker. Two weeks later, a third sort was repeated based on the selection mCherry-expressing cells.

## Characterization of successful knock-in events by PCR and Sanger sequencing

DNA from single-cell-derived bacterial or cell colonies was extracted with a Quick-DNA Miniprep Kit (Zymo #D3025). Combinatorial primer sets designed to recognize the 5′ and 3′ knock-in boundaries were used with the following PCR cycling conditions: 98℃ for two mins, followed by 40 cycles of 98℃ for 30 s and 68℃ for 60 s. The sequences for genotyping primers are provided in *Supplementary file 1*. After electrophoresis, the bands that were at the expected size were cut out, purified, and sequenced with two specific primers (*Supplementary file 1*).

## CRISPR library construction and screening

A set of ~11,000 sgRNA oligos that target 1639 human transcription factors were designed for array-based oligonucleotide synthesis (CustomArray). Unique binding of each sgRNA was verified by sequence blast against the whole human genome. In the sgRNA pooled library, seven gRNAs against each of the 1639 human transcription factors were obtained from validated sgRNA libraries published previously (*Wang et al., 2015*; *Doench et al., 2016*; *Sanjana et al., 2014*; *Ma et al., 2015*; *Tzelepis et al., 2016*; *Hart et al., 2015*; *Hart et al., 2017*; *Smith et al., 2008*; *Park et al., 2017*). The synthesized oligo pool was amplified by PCR and cloned into LentiGuide-Puro backbone (#52963) by in-fusion assembly (Clontech #638909). The *HOXA9$^{P2A-mCherry}$* reporter cell line was

overexpressed with lentiviral Cas9 followed by infection of pooled sgRNA library at low M.O.I (~0.3). Infected cells were selected by blasticidine and puromycin and later sorted for mCherry^High and mCherry^Low populations between days 10–12. The sgRNA sequences were recovered by genomic PCR analysis and deep sequencing using MiSeq for single-end 150 bp read length (Illumina). The primer sequences used for cloning and sequencing are listed in *Supplementary file 1*. The sgRNA sequences are described in *Supplementary file 2*. High-titer lentivirus stocks were generated in 293 T cells as previously described (*Vo et al., 2017*).

## Data analysis of CRISPR screening

The raw FASTQ data were de-barcoded and mapped to the original reference sgRNA library. The differentially enriched sgRNAs were defined by comparing normalized counts between sorted cells in the top 10% and those in the bottom 10% of mCherry-expressing bulk populations. Two independent replicate screenings were performed with the *HOXA9^P2A-mCherry* reporter cell line stably expressing Cas9. Normalized counts for each sgRNA were extracted and used to identify differentially enriched sgRNA by DESeq2 (*Love et al., 2014*). The combined analysis of seven sgRNAs against each human transcription factor was conducted by using the MAGeCK algorithm (*Li et al., 2014*). Detailed screening results were included in *Supplementary file 2*.

## Fluorescence imaging and analysis

0.1% of DMSO (vehicle control) or 10 doses of SGC0946 with a half log scale (0.3 nM-10 μM) were first dispensed into 384-well plates (in quadruplicate, four wells per dose). Suspension-cultured SEM cells were immediately plated into the 384-well plate (20,000 cells / well). Six days after drug treatment, the cells were fixed with 4% paraformaldehyde for 10 mins at room temperature, followed by Hoechst staining for 15 mins at room temperature. Fluorescence images (Hoechst and mCherry) were taken by a CellVoyager 8000 high content imager (Yokogawa). The acquired images were processed by using the Columbus Image Data Storage and Analysis system (Perkin Elmer) to count the number of positive cells and measure fluorescent intensity. To determine the changes of mCherry intensity in SEM expressing *HOXA9^P2A-mCherry*, we measured average mCherry intensity of four fields per well and normalized to vehicle (0.1% DMSO) treated control. Wild-type SEMs with no fluorescence were included as negative controls.

## Cut and Run assay

Cut and Run assay was conducted following the protocol described previously (*Skene and Henikoff, 2017*). In brief, three million cells were collected for each sample. The USF2 antibody (NBP1-92649, Novus) was used at a 1:100 dilution. Library construction was performed using the NEBNext UltraII DNA Library Prep Kit from NEB (E7645S). Indexed samples were run using the Illumina Next-seq 300-cycle kit. Cut and Run raw reads were mapped to genome hg19. by bowtie 2.3.4 with default parameter. The mapping file were converse to. bw file by bamCoverage (*Langmead and Salzberg, 2012*; *Ramírez et al., 2014*).

## Flow cytometry

Suspension-cultured SEM and OCI-AML2 cells were collected by centrifugation at 800X*g*, filtered through a 70 μm filter, and analyzed for mCherry on a BD FACS Aria III flow cytometer with a negative control. The 4,6-diamidino-2-phenylindole (DAPI) staining was conducted prior to sorting to exclude dead cells.

## Inhibitor treatment

SEM and OCI-AML2 cells were seeded at a density of $1 \times 10^5$ cells/mL in medium supplemented with DMSO vehicle or different doses (from 0.5 μM to 15 μM) of the DOT1L inhibitor SGC0946 (MedChemExpress #HY-15650). Medium was replaced every three days, and fresh inhibitor was added. At day-6 post-treatment, cells were collected for flow cytometry analysis and RNA extraction.

## Fluorescence in situ hybridization

An ~800 bp purified *P2A-mCherry* DNA fragment was labeled with a red-dUTP (AF594, Molecular Probes) by nick translation, and a *HOXA9* BAC clone (CH17-412I12/7p15.2) was labeled with a green-dUTP (AF488, Molecular Probes). Both of labeled probes were combined with sheared human DNA and independently hybridized to fix the interphase and metaphase nuclei derived from each sample by using routine cytogenetic methods in a solution containing 50% formamide, 10% dextran sulfate, and 2XSSC. The cells were then stained with DAPI and analyzed.

## Quantitative real-time PCR

Total RNA was collected by using TRIzol (Thermo Fisher Scientific #15596026) or Direct-zol RNA Miniprep Kit (Zymo #R2052). Reverse transcription was performed by using a High-Capacity cDNA Reverse Transcriptase Kit (Applied Biosystems #4374966). Real-time PCR was performed by using FAST SYBR Green Master Mix (Applied Biosystems #4385612) in accordance with the manufacturer's instructions. Relative gene expression was determined by using the ΔΔ-CT method (*Schmittgen and Livak, 2008*). All Q-PCR primers used in this study are listed in *Supplementary file 1*.

## Competitive proliferation assay

For evaluating the impact of USF2 sgRNAs on leukemia expansion, cell cultures were lentivirally transduced with individual USF2 sgRNAs in mCherry expressing vector, followed by measurement of the mCherry-positive percentage at various days post-infection using flow cytometry. The rate of mCherry-positive percentage was normalized to that of Day 3 and declined over time, which was used to infer a defect in cell accumulation conferred by a given sgRNA targeting USF2 relative to the uninfected cells in the same culture.

## Immunoblotting

Cells lysate was prepared by using RIPA buffer followed with SDS-PAGE (Thermo Fisher Scientific) and transferred to a PVDF membrane according to the manufacturer's protocols (Bio-Rad) at constant 100 V for 1 hr. After blocking incubation with 5% non-fat milk in TBS-T (10 mM Tris, pH 8.0, 150 mM NaCl, 0.5% Tween-20) for 1 hr at room temperature, the membrane was incubated with antibodies against GAPDH (Thermo Fisher Scientific, AM4300, 1:10,000), MYC (Cell Signaling Technology, #9402, 1:1,000), USF2 (Novus, NBP1-92649, 1:2,000), USF1 (Proteintech, 22327–1-AP, 1:2,000), GAPDH (Thermo Fisher Scientific, AM4300, 1:10,000), Vinculin (Proteintech, 26520–1-AP, 1:2,000) and CTCF (abcam, ab70303, 1:1,000) at 4°C for 12 hr with gentle shaking. Membranes were washed three times for 30 min and incubated with a 1:2000 dilution of horseradish peroxidase-conjugated anti-mouse or anti-rabbit antibodies for 2 hr at room temperature. Blots were washed with TBS-T three times for 30 min and developed with the ECL system (Amersham Biosciences) according to the manufacturer's protocol.

## Statistics

All values are shown as the mean ± SEM. Statistical analyses were performed with GraphPad Prism software, version 8.0. p-Values were calculated by performing a two-tailed *t*-test.

# Acknowledgements

We gratefully acknowledge the staffs of the Hartwell Sequencing, Cytogenetics, Flow Cytometry and Cell Sorting Shared Resource facility within the Comprehensive Cancer Center of St. Jude Children's Research Hospital. We thank Li and Lu laboratory members for critical comments and discussion. We thank Dr. Cherise Guess for helping with scientific editing.

# Additional information

### Funding

| Funder | Grant reference number | Author |
|--------|------------------------|--------|
| Leukemia Research Founda- | | Rui Lu |

tion

| Concern Foundation | | Rui Lu |
| --- | --- | --- |
| American Cancer Society | IRG15-59-IRG | Rui Lu |
| Young Supporters Board of the O'Neal Comprehensive Cancer Center | | Rui Lu |
| National Cancer Institute | P30CA021765-37 | Chunliang Li |
| National Heart, Lung, and Blood Institute | R01HL153220 | Yang Zhou |

The funders had no role in study design, data collection and interpretation, or the decision to submit the work for publication.

## Author contributions
Hao Zhang, Data curation, Formal analysis, Validation, Investigation, Methodology, Writing - original draft, Writing - review and editing; Yang Zhang, Data curation, Software, Formal analysis, Validation, Investigation, Methodology, Writing - original draft, Writing - review and editing; Xinyue Zhou, Shaela Wright, Formal analysis, Investigation, Methodology; Judith Hyle, Formal analysis, Validation, Investigation, Methodology, Writing - original draft, Writing - review and editing; Lianzhong Zhao, Jie An, Hyeong-Min Lee, Investigation, Methodology; Xujie Zhao, Ying Shao, Yang Zhou, Methodology; Beisi Xu, Software, Formal analysis; Taosheng Chen, Supervision, Methodology; Xiang Chen, Software; Rui Lu, Chunliang Li, Conceptualization, Resources, Formal analysis, Supervision, Funding acquisition, Validation, Investigation, Visualization, Methodology, Writing - original draft, Project administration, Writing - review and editing

## Author ORCIDs
Yang Zhang ⬤ https://orcid.org/0000-0003-1712-7211
Beisi Xu ⬤ http://orcid.org/0000-0003-0099-858X
Hyeong-Min Lee ⬤ http://orcid.org/0000-0002-9381-3611
Taosheng Chen ⬤ http://orcid.org/0000-0001-6420-3809
Rui Lu ⬤ https://orcid.org/0000-0003-1593-2612
Chunliang Li ⬤ https://orcid.org/0000-0002-5938-5510

## Decision letter and Author response
Decision letter https://doi.org/10.7554/eLife.57858.sa1
Author response https://doi.org/10.7554/eLife.57858.sa2

# Additional files

## Supplementary files
• Supplementary file 1. Oligo information used in this paper.
• Supplementary file 2. Raw count information related to CRISPR screen.
• Transparent reporting form

## Data availability
All plasmids created in this study will be deposited to Addgene. Raw data collected from Cut&Run were deposited at NCBI GEO (GSE140664). Raw data collected from CRISPR screening were included in Supplementary File 2. Publicly available dataset used in this study were cited accordingly including Figures 1E and S5D: GSE120781; Figure 1-supplement 1A-C: GSE13159; Figure 3-supplement 2C: GSE126619, GSE74812, GSE89485; Figure 3-supplement 3A: ENCODE (HCT116); Figure 5-supplement 3A-C: European Genome-phenome Archive (EGA) under accession number EGAS00001003266, EGAS00001000654, EGAS00001001952, EGAS00001001923, EGAS00001002217 and EGAS00001000447.

The following dataset was generated:

| Author(s) | Year | Dataset title | Dataset URL | Database and Identifier |
|---|---|---|---|---|
| Zhang Y, Li C | 2020 | Functional Interrogation of HOXA9 Regulome in Leukemia via Endogenous Reporter-based CRISPR/Cas9 screen. | https://www.ncbi.nlm.nih.gov/geo/query/acc.cgi?acc=GSE140664 | NCBI Gene Expression Omnibus, GSE140664 |

The following previously published datasets were used:

| Author(s) | Year | Dataset title | Dataset URL | Database and Identifier |
|---|---|---|---|---|
| Hyle J, Zhang Y, Wright S, Xu B, Shao Y, Easton J, Tian L, Feng R, Xu P, Li C | 2019 | Acute deletion of CTCF disrupted enhancer-promoter regulation of MYC in human cancer cells | https://www.ncbi.nlm.nih.gov/geo/query/acc.cgi?acc=GSE120781 | NCBI Gene Expression Omnibus, GSE120781 |
| Kohlmann A, Kipps TJ, Rassenti LZ, Downing JR, Shurtleff SA, Mills KI, Gilkes AF, Hofmann W-K, Basso G, Dell'orto MC, Foà R, Chiaretti S, Vos JD, Rauhut S, Papenhausen PR, Hernández JM, Lumbreras E, Yeoh AE, Koay ES, Li R, Liu W-M, Williams PM, Wieczorek L, Haferlach T | 2008 | Microarray Innovations in LEukemia (MILE) study: Stage 1 data | https://www.ncbi.nlm.nih.gov/geo/query/acc.cgi?acc=GSE13159 | NCBI Gene Expression Omnibus, GSE13159 |
| Hyle J, Zhang Y, Wright S, Xu B, Shao Y, easton J, Tian L, Feng R, Xu P, Li C | 2019 | Acute deletion of CTCF disrupted enhancer-promoter regulation of MYC in human cancer cells | https://www.ncbi.nlm.nih.gov/geo/query/acc.cgi?acc=GSE126619 | NCBI Gene Expression Omnibus, GSE126619 |
| Kerry J, Milne TA | 2015 | MLL-rearranged acute lymphoblastic leukemias upregulate BCL-2 through H3K79 methylation and are highly sensitive to the BCL-2 specific antagonist ABT-199 | https://www.ncbi.nlm.nih.gov/geo/query/acc.cgi?acc=GSE74812 | NCBI Gene Expression Omnibus, GSE74812 |
| Liang K, Smith ER, Shilatifard A | 2017 | Therapeutic targeting MLL degradation pathways in MLL-rearranged leukemia | https://www.ncbi.nlm.nih.gov/geo/query/acc.cgi?acc=GSE89485 | NCBI Gene Expression Omnibus, GSE89485 |
| Gu Z, Churchman ML, Roberts KG, Moore I, Zhou X, Nakitandwe J, Hagiwara K, Pelletier S, Gingras S, Berns H, Payne-Turner D, Hill A, Iacobucci I, Shi L, Pounds S, Cheng C, Pei D, Qu C, Newman S, Devidas M, Dai Y, Reshmi SC, Gastier-Foster J, Raetz EA, Borowitz MJ, Wood BL, Carroll WL, Zweidler-McKay PA, Rabin KR, Mattano LA, Maloney KW, Rambaldi A, Spinelli O, Radich JP, | 2019 | PAX5-driven Subtypes of B-cell Acute Lymphoblastic Leukemia | https://www.ebi.ac.uk/ega/studies/EGAS00001003266?order=type&sort=asc | European Genome-phenome Archive, EGAS00001003266 |

| | | | | |
|---|---|---|---|---|
| Minden MD, Rowe JM, Luger S, Litzow MR, Tallman MS, Racevskis J, Zhang Y, Bhatia R, Kohlschmidt J, Mrózek K, Bloomfield CD, Stock W, Kornblau S, Kantarjian HM, Konopleva M, Evans WE, Jeha S, Pui CH, Yang J, Paietta E, Downing JR, Relling MV, Zhang J, Loh ML, Hunger SP, Mullighan CG | | | | |
| Gu Z, Churchman ML, Roberts KG, Moore I, Zhou X, Nakitandwe J, Hagiwara K, Pelletier S, Gingras S, Berns H, Payne-Turner D, Hill A, Iacobucci I, Shi L, Pounds S, Cheng C, Pei D, Qu C, Newman S, Devidas M, Dai Y, Reshmi SC, Gastier-Foster J, Raetz EA, Borowitz MJ, Wood BL, Carroll WL, Zweidler-McKay PA, Rabin KR, Mattano LA, Maloney KW, Rambaldi A, Spinelli O, Radich JP, Minden MD, Rowe JM, Luger S, Litzow MR, Tallman MS, Racevskis J, Zhang Y, Bhatia R, Kohlschmidt J, Mrózek K, Bloomfield CD, Stock W, Kornblau S, Kantarjian HM, Konopleva M, Evans WE, Jeha S, Pui CH, Yang J, Paietta E, Downing JR, Relling MV, Zhang J, Loh ML, Hunger SP, Mullighan CG | 2019 | PAX5-driven Subtypes of B-cell Acute Lymphoblastic Leukemia | https://www.ebi.ac.uk/ega/studies/EGAS00001000654 | European Genome-phenome Archive, EGAS00001000654 |
| Gu Z, Churchman ML, Roberts KG, Moore I, Zhou X, Nakitandwe J, Hagiwara K, Pelletier S, Gingras S, Berns H, Payne-Turner D, Hill A, Iacobucci I, Shi L, Pounds S, Cheng C, Pei D, Qu C, Newman S, Devidas M, Dai Y, Reshmi SC, Gastier-Foster J, Raetz EA, Borowitz MJ, Wood BL, Carroll WL, Zweidler-McKay PA, Rabin KR, Mattano | 2019 | PAX5-driven Subtypes of B-cell Acute Lymphoblastic Leukemia | https://www.ebi.ac.uk/ega/studies/EGAS00001001952 | European Genome-phenome Archive, EGAS00001001952 |

| | | | | |
|---|---|---|---|---|
| LA, Maloney KW, Rambaldi A, Spinelli O, Radich JP, Minden MD, Rowe JM, Luger S, Litzow MR, Tallman MS, Racevskis J, Zhang Y, Bhatia R, Kohlschmidt J, Mrózek K, Bloomfield CD, Stock W, Kornblau S, Kantarjian HM, Konopleva M, Evans WE, Jeha S, Pui CH, Yang J, Paietta E, Downing JR, Relling MV, Zhang J, Loh ML, Hunger SP, Mullighan CG | | | | |
| Gu Z, Churchman ML, Roberts KG, Moore I, Zhou X, Nakitandwe J, Hagiwara K, Pelletier S, Gingras S, Berns H, Payne-Turner D, Hill A, Iacobucci I, Shi L, Pounds S, Cheng C, Pei D, Qu C, Newman S, Devidas M, Dai Y, Reshmi SC, Gastier-Foster J, Raetz EA, Borowitz MJ, Wood BL, Carroll WL, Zweidler-McKay PA, Rabin KR, Mattano LA, Maloney KW, Rambaldi A, Spinelli O, Radich JP, Minden MD, Rowe JM, Luger S, Litzow MR, Tallman MS, Racevskis J, Zhang Y, Bhatia R, Kohlschmidt J, Mrózek K, Bloomfield CD, Stock W, Kornblau S, Kantarjian HM, Konopleva M, Evans WE, Jeha S, Pui CH, Yang J, Paietta E, Downing JR, Relling MV, Zhang J, Loh ML, Hunger SP, Mullighan CG | 2019 | PAX5-driven Subtypes of B-cell Acute Lymphoblastic Leukemia | https://www.ebi.ac.uk/ega/studies/EGAS00001001923 | European Genome-phenome Archive, EGAS00001001923 |
| Gu Z, Churchman ML, Roberts KG, Moore I, Zhou X, Nakitandwe J, Hagiwara K, Pelletier S, Gingras S, Berns H, Payne-Turner D, Hill A, Iacobucci I, Shi L, Pounds S, Cheng C, Pei D, Qu C, Newman S, Devidas M, Dai Y, Reshmi SC, Gastier-Foster J, Raetz EA, Borowitz MJ, Wood | 2019 | PAX5-driven Subtypes of B-cell Acute Lymphoblastic Leukemia | https://www.ebi.ac.uk/ega/studies/EGAS00001002217 | European Genome-phenome Archive, EGAS00001002217 |

| | | | | |
|---|---|---|---|---|
| BL, Carroll WL, Zweidler-McKay PA, Rabin KR, Mattano LA, Maloney KW, Rambaldi A, Spinelli O, Radich JP, Minden MD, Rowe JM, Luger S, Litzow MR, Tallman MS, Racevskis J, Zhang Y, Bhatia R, Kohlschmidt J, Mrózek K, Bloomfield CD, Stock W, Kornblau S, Kantarjian HM, Konopleva M, Evans WE, Jeha S, Pui CH, Yang J, Paietta E, Downing JR, Relling MV, Zhang J, Loh ML, Hunger SP, Mullighan CG | | | | |
| Gu Z, Churchman ML, Roberts KG, Moore I, Zhou X, Nakitandwe J, Hagiwara K, Pelletier S, Gingras S, Berns H, Payne-Turner D, Hill A, Iacobucci I, Shi L, Pounds S, Cheng C, Pei D, Qu C, Newman S, Devidas M, Dai Y, Reshmi SC, Gastier-Foster J, Raetz EA, Borowitz MJ, Wood BL, Carroll WL, Zweidler-McKay PA, Rabin KR, Mattano LA, Maloney KW, Rambaldi A, Spinelli O, Radich JP, Minden MD, Rowe JM, Luger S, Litzow MR, Tallman MS, Racevskis J, Zhang Y, Bhatia R, Kohlschmidt J, Mrózek K, Bloomfield CD, Stock W, Kornblau S, Kantarjian HM, Konopleva M, Evans WE, Jeha S, Pui CH, Yang J, Paietta E, Downing JR, Relling MV, Zhang J, Loh ML, Hunger SP, Mullighan CG | 2019 | PAX5-driven Subtypes of B-cell Acute Lymphoblastic Leukemia | https://www.ebi.ac.uk/ega/studies/EGAS00001000447 | European Genome-phenome Archive, EGAS00001000447 |

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
