## [Decision Letter]

[Editors' note: this paper was reviewed by Review Commons.]

**Acceptance summary:**

Aberrant *HOXA9* expression is a hallmark of most aggressive acute leukemias, thus understanding the regulation of *HOXA9* gene in leukemias is important. By utilizing CRISPR/Cas9 mediated screens in a reporter cell line, this study identifies new regulators controlling *HOXA9* expression, which are subsequently validated in multiple cell lines. This study provides the foundation for developing therapeutic strategies, and the *HOXA9^P2A-mCherry^* reporter is a useful tool to the leukemia and epigenetic fields.

**Decision letter after peer review:**

Thank you for submitting your article "Functional Interrogation of *HOXA9* Regulome in MLLr Leukemia via Reporter-based CRISPR/Cas9 screen" for consideration by *eLife*. Your article has been reviewed by three peer reviewers, and the evaluation has been overseen by a Reviewing Editor and Maureen Murphy as the Senior Editor. The reviewers have discussed the reviews with one another and the Reviewing Editor has drafted this decision to help you prepare a revised submission.

As the editors have judged that your manuscript is of interest, but as described below that additional experiments are required before it is published, we would like to draw your attention to changes in our revision policy that we have made in response to COVID-19 (https://elifesciences.org/articles/57162). First, because many researchers have temporarily lost access to the labs, we will give authors as much time as they need to submit revised manuscripts. We are also offering, if you choose, to post the manuscript to bioRxiv (if it is not already there) along with this decision letter and a formal designation that the manuscript is 'in revision at *eLife*'. Please let us know if you would like to pursue this option. (If your work is more suitable for medRxiv, you will need to post the preprint yourself, as the mechanisms for us to do so are still in development.)

Summary:

In this manuscript, the authors study the transcriptional regulation of *HOXA9*, a transcription factor that plays a central role in homeostasis of immature hematopoietic cell types and in the development of leukemia. They use the CRISPR/Cas9 technique to introduce a fluorescence reporter cassette into the endogenous *HOXA9* locus of a human MLL/AF4-rearranged B-ALL cell line. After validating this engineered cell line, they perform multiple genetic screens to identify potential transcriptional regulators of *HOXA9* and to delineate essential transcription factors in this cell line. They identify USF2 as new transcription factor that modulates expression of *HOXA9*.

Major revisions:

If the authors can commit to adding the following data, as they indicate in their rebuttal, the manuscript would be greatly strengthened and could be considered acceptable:

1) The authors should include their data on the independent loss-of-function CRISPR transcription factor screen in SEM *HOXA9^P2A-mCherry^* MLLr reporter line ectopically expressing HOXA9-MEIS1 to overcome the possibility that key regulators could be missed in the CRISPR/Cas9 screen due to survival dropout.

2) The authors should include supporting data for the key observations in the manuscript in other cell lines; for example, as indicated by the authors, the data gathered using an additional *HOXA9* MLLr AML reporter cell line established in OCI-AML2 cells to further support findings from the initial in SEM MLLr ALL reporter line.

3) The authors should perform USF2 knockout experiments in multiple non-MLLr cell lines according to the reviewer's suggestions. As an example, the authors should repeat the competitive proliferation assay to determine the effects of the single knockout of USF1 and USF2 vs the double knockout in SEM cells and other MLLr leukemia cell lines with proper controls.

---

## [Author Response]

Major revisions:If the authors can commit to adding the following data, as they indicate in their rebuttal, the manuscript would be greatly strengthened and could be considered acceptable:1) The authors should include their data on the independent loss-of-function CRISPR transcription factor screen in SEM HOXA9-P2A-mCherry MLLr reporter line ectopically expressing HOXA9-MEIS1 to overcome the possibility that key regulators could be missed in the CRISPR/Cas9 screen due to survival dropout.

Thank you for the reviewer’s positive comments. This reviewer raised an important question that key upstream regulators of *HOXA9* could be missed due to a survival disadvantage. To mitigate this challenge, we have conducted an independent CRISPR/Cas9 TF screen in *HOXA9^P2A-mCherry^* reporter SEM cells ectopically expressing *HOXA9* together with its functional partner MEIS1. In this regard, exogenously expressed *HOXA9* could rescue the potential cell loss due to decreased *HOXA9* expression in SEM cells, while the level of endogenous *HOXA9* is still monitored by reporter. As a result, we have identified more well-known regulators of *HOXA9* which are also considered as survival essential genes. Among the top 10 hits from this screen, DOT1L and *HOXA9* were enriched. KMT2A, the translocation partner of MLL-AF4 in SEM cells, was identified in the HOXA9-MEIS1 rescue TF screen but not the original screen without ectopic expression of *HOXA9*. Notably, the MYST acetyltransferase HBO1 (also known as KAT7 or MYST2) and several members of the HBO1 protein complex, which were recently shown as critical regulators of leukemia stem cell maintenance, were also identified among the top hits (MacPherson et al., 2020). Most importantly, USF2 was still among the top hits in this screen. Based on these observations, we believe that our reporter-based screen is sensitive to identify *HOXA9* regulators. We have added these new data to Figures 3B, 3D-3F, Figure 3—figure supplement 1.

2) The authors should include supporting data for the key observations in the manuscript in other cell lines; for example, as indicated by the authors, the data gathered using an additional HOXA9 MLLr AML reporter cell line established in OCI-AML2 cells to further support findings from the initial in SEM MLLr ALL reporter line.

Thank you for the reviewer’s comments. First, we successfully established an additional *HOXA9^P2A-mCherry^* MLLr reporter line from human AML OCI-AML2 cells using the same CHASE-knock-in protocol. The OCI-AML2^HOXA9-P2A-mCherry^ cell line has been fully characterized by *HOXA9* transcriptional response of the reporter allele upon genetic perturbation and pharmaceutical inhibition of known *HOXA9* regulators. This result suggests that our strategy for generating *HOXA9* reporter lines are highly reproducible and can be applied into a broad range of cell lines. Second, we used MLLr AML cell lines OCI-AML2, MOLM13 and NOMO-1 to confirm the effects of USF1 and USF2 single and double knockout on *HOXA9* expression and cell proliferation. We have added these new data to Figure 2—figure supplement 1, Figures 5I-5J and Figure 5—figure supplement 1.

3) The authors should perform USF2 knockout experiments in multiple non-MLLr cell lines according to the reviewer's suggestions. As an example, the authors should repeat the competitive proliferation assay to determine the effects of the single knockout of USF1 and USF2 vs the double knockout in SEM cells and other MLLr leukemia cell lines with proper controls.

Thank you for the reviewer’s comments. We have performed USF2 knockout experiments in two non-MLLr cell lines U937 and OCI-AML3, which also expressed *HOXA9*. USF2 was completely depleted by two sgRNAs in these two cell lines shown by immunoblotting. However, *HOXA9* mRNA expression was not reduced upon USF2 loss. Also, we have conducted competitive proliferation assay in SEM, OCI-AML2 and MOLM13 cells to determine the effects of single knockout of USF1, USF2 and double knockout. As suggested, in the new experiments, we have included two negative controls: (1) a non-targeting control (sgRNA against Luciferase gene, which would not bind to genomic RNA nor induce double strand DNA break); (2) a control guide RNA targeting the human endogenous gene ROSA26. New data are shown as Figure 5G-5J and Figure 5—figure supplement 1.

Reviewer #1 (Evidence, reproducibility and clarity):In this study, the authors make a HOXA9^P2A-mCherry^ fusion in the t(4;11) cell line SEM and use this line to perform a CRISPR/Cas9 deletion screen to identify novel regulators of HOXA9 expression. The top ranked candidate was USF2. They used Cut&Run to show USF2 binds directly to the HOXA9 gene and showed that guide RNAs specific for USF2 downregulate HOXA9 expression. They then performed a CRISPR/Cas9 dropout screen in SEM cells and independently verified that USF2 is a key survival target in these cells. They finish by performing a Cas9-KRAB screen using guides across the HOXA9 locus to identify putative new regulatory regions.Major points:1) The authors start off by making a very good case for better understanding upstream regulators of the HOXA9 gene, as it is overexpressed in a wide range of different leukemias. However, the entire main screen is performed in the MLLAF4 driven cell line SEM. The HOXA9 gene is a direct target of MLL-AF4 regulation, thus it is not clear if this is the best system to identify key HOXA9 regulators that are also applicable to other contexts. Ideally, it would be best to have done the screening in a non-MLL rearranged (MLLr) cell line. At the very least, USF2 knockouts should be performed in multiple non-MLLr cell lines to determine if this is an MLLr-specific TF or a more widely important TF needed for HOXA9 regulation.2) Could a more refined deletion analysis of the putative regulatory regions identified in Figure 8 be performed in SEM cells exogenously expressing HOXA9 (so the cells are not killed by the deletion analysis)? It would be interesting to see if any other TF motifs in this region correlate with TFs identified in either the mCherry expression screen (Figure 3B) or the dropout screen (Figure 6D).Reviewer #1 (Significance):The authors are correct that HOXA9 is a generally important target in leukemia and little is understood about its upstream regulators. With USF2 identified as a key upstream TF, this work represents an important attempt to increase our understanding in that area. I have published multiple papers on gene regulation in MLL leukemias and have a specific focus on MLL-AF4 driven B-ALL.

We appreciate the positive comments.

Referees cross commenting:I agree with the points reviewer #2 has raised. I think reviewer #3 in particular has several of the same concerns as I do, and I agree with all of their major comments. They also raise the important point that the way the screen was performed, key upstream regulators of HOXA9 could be missed.

Thank you for the reviewer’s positive comments. This reviewer raised an important question that key upstream regulators of *HOXA9* could be missed due to a survival disadvantage. To mitigate this challenge, we have conducted an independent CRISPR/Cas9 TF screen in *HOXA9^P2A-mCherry^* reporter SEM cells ectopically expressing the *HOXA9*, together with its functional partner MEIS1. In this regard, exogenously expressed *HOXA9* could rescue the potential cell loss due to decreased *HOXA9* expression in SEM cells, while the level of endogenous *HOXA9* is still monitored by mCherry reporter. As a result, we have identified more well-known regulators of *HOXA9* which are also considered as survival essential genes. Among the top 10 hits from this screen, DOT1L and *HOXA9* were enriched. KMT2A, the translocation partner of MLL-AF4 in SEM cells, was identified in the HOXA9-MEIS1 rescue TF screen but not the original screen without ectopic expression of *HOXA9*. Notably, the MYST acetyltransferase HBO1 (also known as KAT7 or MYST2) and several members of the HBO1 protein complex, which were recently shown as critical regulators of leukemia stem cell maintenance, were also identified among the top hits (MacPherson et al., 2020). Most importantly, USF2 was still among the top hits in this screen. Based on these observations, we believe that our reporter-based screen is sensitive to identify *HOXA9* regulators. We add these new data to Figures 3B, 3D-3F, Figures 3—figure supplement 1D-F. However, some genes may experience poor sgRNA targeting efficiency leading to the possibility that important regulators could be overlooked, which is a known limitation of all CRISPR/Cas9 library screens.

I also just want to clarify something I don't think I made clear enough in my original review. Using an MLL rearranged cell line is not the best way to identify key, specific HOXA9 regulators. Since HOXA9 is commonly an MLL fusion protein (MLL-FP) activated target, MLL-FP activated expression may be dominant in the system, masking the effect of more highly specific transcription factors. This makes it more likely that such a screen would simply identify factors that are part of the general transcription activation machinery (e.g. SP1 or components of the pre-initiation complex). I don't know a lot about USF1 or 2, but I suspect it may fall into this sort of general transcription activation category. Thus, it will be key to know both whether i) this factor is important in other cell types and also ii) how generally required it is for transcriptional activation at other gene targets.

Thank you for the reviewer’s comments. We agree with the reviewer that *HOXA9* is a direct target of MLL-FP, and MLL-FP plays a key role in *HOXA9* regulation in MLL-rearranged cell lines. In fact, in addition to MLL-FP, a list of other regulators has been identified to be key regulators of *HOXA9* in this particular class of genetic alterations including DOT1L, *ENL*, ASHL1 and recently the HBO1 complex. However, it remains unknown whether additional regulators are involved in regulating *HOXA9*. This study was designed to specifically identify potential regulators of *HOXA9* in an MLLr background, which has not been thoroughly investigated yet. Generally, most human cancer cells do not express elevated levels of *HOXA9*. Significantly high levels of *HOXA9* have been observed in MLLr, NPM1c, and NUP98-HOXA9 fusion leukemia subtypes, among which MLLr is considered a high-risk leukemia with poor diagnosis and outcome. We chose MLLr cell lines because MLL-FP complex is well studied, and many know regulators can serve as positive controls for validating our screening while allowing us to discover possible new hits. In addition, the pipeline described in this manuscript can be easily transferred to other AML lines for studying *HOXA9* regulation in the future. While USF1 and USF2 have been characterized as general transcription factors, we found that they positively regulate transcription levels of *HOXA9* in the MLLr leukemia cell lines SEM, OCI-AML2, NOMO-1 and MOLM13. USF2 knockout in the non-MLLr U937 and OCI-AML3 cell line did not affect *HOXA9* expression nor cell survival, indicating its regulation of *HOXA9* could be exerted in a cell type-specific manner.

Reviewer #2 (Evidence, reproducibility and clarity):In this manuscript, Hao Zhang et al. generate a HOXA9-mCherry Knock-in reporter into an MLL-rearranged B-ALL cell line, SEM. By using this new reporter cell line, they apply CRISPR/Cas9 screen to identify a novel transcriptional factor USF2/USF1 which binds at HOXA9 promoter and controls HOXA9 expression at transcriptional level in B-ALL. The major scientific conclusion is convincing. Furthermore, the reliability and reproducibility of the experiments are decent.

Thank you for the reviewer’s positive comments.

However, one of the major claims that USF2 and HOXA9 were positively correlated in MLL-rearranged B-ALL patients is preliminary. The R score is weak especially it is calculated upon separation of HOXA9 high and low expression group.

Correlation analysis between USF2 and *HOXA9* was unbiasedly conducted in the largest transcriptomic cohort of human B-ALL patients collected in our institution by Dr. Charles Mullighan (Nat Genet. 2019 Feb;51(2):296-307). As a positive control, transcriptional correlation of USF1 and USF2 was consistently detected in all patients. However, significant positive correlation between USF2 and *HOXA9* was specifically identified only in patients with MLLr (136 cases) subtype. Due to the limitation of patient materials, we could not conduct additional experiments on human samples to further extend the study. We have now moved these data to Figure 5—figure supplement 3.

Major comments:1) Figure 1—figure supplement A and C: the expression of HOXA9 in Pro-B-ALL t(11q23)/MLL in panel A is relatively high. However, the panel C showed contradictory results as no HOXA9 OE is found in Pro-B-ALL t(11q23)/MLL as well as the expression is relatively low. Please discuss why this might be the case.

We apologize for the mis-labeling of leukemia subtypes. The correct figure (Figure 1—figure supplement 1C) is provided to replace original Figure S1C. It’s clear that Pro-B-ALL t(11q23)/MLL patients demonstrated high levels of *HOXA9*.

2) Figure 2 A-D: the results shown in the figure are composed of one sgRNA for each gene and in one cell line per leukemia type. The authors need to add an additional sgRNA and one additional cell line per leukemia type.

Thank you for the reviewer’s comments. In this figure, we are trying to confirm the response of the *HOXA9^P2A-mCherry^* knockin reporter upon known transcriptional or pharmaceutical inhibitors of *HOXA9*. The DOT1L sgRNA and *ENL* sgRNA used in Figures 2A-2D were previously validated and well characterized in other studies. Therefore, we felt confident these controls could be used to faithfully validate the response of the *HOXA9* reporter. In addition, the CRISPR/Cas9 screen, which used seven sgRNAs per target, further confirmed the results observed in Figures 2A-2D. We have also generated an additional *HOXA9^P2A-mCherry^* reporter cell line using the MLLr AML cell line OCI-AML2 and comprehensively characterized the knockin allele. We have added these new data to Figure 2—figure supplement 1.

3) Figure 3—figure supplement 3B: Please document CTCF expression upon siRNA mediated CTCF. Furthermore, the Y axis description of the figure is missing.

Thank you for the reviewer’s comments. We have provided the immunoblotting and Q-PCR results showing that CTCF expression was completely inhibited by siRNAs. The Y axis description has been added. See Figure 3—figure supplement 3.

4) Figure 5C: Please report expression of HOXA9.

Thank you for the reviewer’s comments. We have tested four commercial *HOXA9* antibodies to detect *HOXA9* protein levels and failed to distinguish *HOXA9* from other homologs, which is a common problem for the entire field. Therefore, we have removed the *HOXA9* immunoblotting result and provided the mRNA expression level of *HOXA9* in Figure 5C.

5) Figure 6D: How/where is HOXA9 expression ranked in this CRISPR screen?

Thank you for the reviewer’s comments. *HOXA9* was ranked at 197^th^ (log_2_FCday12/day0=-0.38; p value=0.069). Although the MeGACK trend suggested *HOXA9* is an essential gene, it’s not among the top 50 essential genes. We reasoned that it may be due to the fact that day12/day0 is not the best timing for investigating the depletion of *HOXA9* sgRNA targeted cells. In addition, compensation from other active *HOXA* genes might play a role upon *HOXA9* loss in our dropout CRISPR screen, which was seen from previous studies (Mol Cell Biol, 26 (10), 3902-3916, 2006; Collins and Hess, 2016). We then designed a sgRNA targeting on TSS of *HOXA9* and delivered the sgRNA to Cas9-expressing SEM cells. Significant reduction of *HOXA9* and proliferation were observed.

6) Figure 6E-F: How to define HOXA9 high expression or low expression? What is the criteria and cut-off to separate the high and low expression of HOXA9 group? This piece of data isn't strong enough to claim the correlation between USF2 and HOXA9 in MLL-rearranged B-ALL.

Thank you for the reviewer’s comments. According to the transcriptome profiling of 136 MLLr B-ALL patients (Gu et al., 2019), two distinct groups of patients were identified and separated by the cutoff of normalized log_2_ expression at 6. We observed the higher correlation between USF2 and *HOXA9^high^* group (pearson’s *r*=0.40, p=9.9e-09) than the correlation from all of the 136 patients (pearson’s *r*=0.25, p=0.004). Due to the unavailability of additional patient materials, we could not conduct further experiments. Instead, we have moved this piece of data to Figure 5—figure supplement 3.

7) Figure 7: Data presented here are not strong enough to claim USF1 and USF2 synthetically regulate HOXA9 expression in MLLr leukemia. Figure A-C please at least use another sg RNA against USF2 to prove HOXA9 expression is downregulated upon CRISPR targeting USF2 in MOLM13 cells. How does USF2 affect cell proliferation in MOML13 cells?

Thank you for the reviewer’s comments. As suggested by the reviewer, we have repeated this experiment in MOLM13 using single USF1, USF2 knockout along with double knockout. We have also investigated the impact on cell proliferation using a competitive proliferation assay with proper controls. We add these new data to Figure 5—figure supplement 1F-G.

8) Supplemental figure legend is missing.

We apologize for this confusion. The supplemental figure legend was provided at the end of the manuscript text in the initial submission.

Reviewer #2 (Significance):Significance for the leukemia and transcription field is high. Overall, Hao Zhang et al. successfully utilize CRISPR/Cas9 technology to study HOXA9 regulation in MLL-rearranged B-ALL. This study helps us understanding the molecular regulation network in HOXA9-driven leukemia.

Thank you for the reviewer’s positive comments.

Reviewer #3 (Evidence, reproducibility and clarity):Summary:In this manuscript, Zhang and colleagues study the transcriptional regulation of HOXA9, a transcription factor that plays a central role in homeostasis of immature hematopoietic cell types and in the development of leukemia. They use the CRISPR/Cas9 technique to introduce a fluorescence reporter cassette into the endogenous HOXA9 locus of a human MLL/AF4-rearranged B-ALL cell line. After validating this engineered cell line, they perform multiple genetic screens to identify potential transcriptional regulators of HOXA9 and to delineate essential transcription factors in this cell line. They identify USF2 as new transcription factor that modulates expression of HOXA9 in MLL-rearranged B-ALL.Major comments:Although the identification of USF2 as a new transcriptional regulator of HOXA9 is interesting, most of the experiments were performed in the SEM cell line, and only few observations were validated in other cell lines. No primary cells/mouse models were used to further validate the role of USF2 in MLLr leukemia. Therefore, the broad validity of the proposed concept is questionable.

Thank you for the reviewer’s positive comments. We have performed USF2 knockout experiments in two non-MLLr cell lines U937 and OCI-AML3, which also expressed *HOXA9*. USF2 was completely depleted by two sgRNAs in these two cell lines shown by immunoblotting. However, *HOXA9* mRNA expression was not reduced upon USF2 loss. Also, we have conducted the competitive proliferation assay in the MLLr cell lines SEM, OCI-AML2 and MOLM13 to determine the effects of single knockout of USF1, USF2 and double knockout. We have added these new data to Figures 5G-5J and Figure 5—figure supplement 1. We think these additional data helped to extend the broad validity of the proposed concept.

Another limitation of the study is inherent to the experimental setup that has been chosen to identify transcriptional regulators of HOXA9. In the screening protocol employed by the authors, transduced cells were flow sorted based on the expression of the fluorescent reporter 10-12 days after transduction of the sgRNA library. With this approach, essential transcription factors that regulate HOXA9 are missed, as cell populations representative in which these factors were mutationally inactivated will most likely be depleted at this timepoint.

This reviewer raised an important question that key upstream regulators of *HOXA9* could be missed due to a survival disadvantage. To mitigate this challenge, we have conducted an independent CRISPR/Cas9 TF screen in *HOXA9^P2A-mCherry^*reporter SEM cells ectopically expressing the HOXA9-MEIS1 fusion protein. Many of the key well-known regulators of *HOXA9* were among the top 10 hits identified from this screen, including KMT2A, DOT1L and *HOXA9*. Notably, the MYST acetyltransferase HBO1 (also known as KAT7 or MYST2) and several members of the HBO1 protein complex, which are known to be critical regulators of leukemia stem cell maintenance, were also identified among the top hits (MacPherson et al., 2020). Most importantly, USF2 was still among the top hits in this screen. Based on these observations, we believe that our reporter-based screen is sensitive to identify *HOXA9* regulators. However, we could not exclude the possibility that some genes may experience poor sgRNA targeting efficiency leading to the possibility that some important regulators could be overlooked, which is a limitation of all CRISPR/Cas9 library screens.

In Figure 2, only one sgRNA for each target was used. The authors should include a second sgRNA targeting DOT1L and ENL, at least in panels A-D. Moreover, HOXA9 expression is only reduced to 40-50% upon knockout of DOT1L or ENL or DOT1L inhibition in Figure 2 B-D. Is this the maximum that can be achieved in this system? If yes, why?

Thank you for the reviewer’s comments. In this figure, we are trying to confirm the response of the *HOXA9^P2A-mCherry^* knockin reporter upon known transcriptional or pharmaceutical inhibitors of *HOXA9*. The DOT1L sgRNA and *ENL* sgRNA used in Figures 2A-2D were previously validated and well characterized in other studies. Therefore, we felt confident these controls could be used to faithfully validate the response of the *HOXA9* reporter. In addition, the CRISPR/Cas9 screen, which used seven sgRNAs per target, further confirmed the results observed in Figures 2A-2D. We have also generated an additional *HOXA9* reporter cell line using the MLLr AML cell line OCI-AML2, which was characterized using the same methods. In our hands, the maximum reduction efficiency of *HOXA9* expression is 40-50% upon sgRNA targeting of DOT1L or *ENL*, or DOT1L inhibition in SEM cells, which is consistent with previous reports. In our second reporter line derived from MLLr AML cell line OCI-AML2, we observed complete repression of *HOXA9* via DOT1L inhibition, indicating the effect of DOT1L inhibition could vary based on cell type by mechanisms that have yet to be identified.

Figure 3D: Show the distribution of DOT1L-targeting sgRNAs.

Thank you for the reviewer’s comments. The distribution of sgRNAs targeting DOT1L was shown along with NT, *HOXA9* and USF2.

Figure 4 is not central to main conclusions of this study. In fact, it mainly provides negative data that might distract readers. The authors might consider moving this figure to the supplementary data. Moreover, the Materials and methods section lacks a description of experiments involving auxin-induced CTCF degradation.

Thank you for the reviewer’s comments. Our data collected from MLLr B-ALL SEM cells targeted by siRNA or CTCF-AID mediated degradation supported the observation from Dr. David Spencer’s lab (Ghasemi et al., 2020), which indicated that *HOXA* gene expression was maintained in the CTCF deletion mutants and transcriptional activity at the *HOXA* locus in NPM1-mutant AML cells does not require long-range CTCF-mediated chromatin interactions. Our CRISPR/screen and degron-mediated CTCF degradation data provided additional insights of CTCF’s function in leukemia cells. We also added detailed description of experiments involving auxin-induced CTCF degradation. We agree with the reviewer that this piece of data might distract readers. Therefore, we move the entire figure to Figure 3—figure supplement 2.

Throughout the paper, the authors use sgRNAs targeting Luciferase (sgLuc) and in one experiment the Rosa locus (in human cells) as negative controls. A better control would have been to target the AAVS1 locus in human cells, as this enables proper control for the DNA damage that is induced by sgRNA-induced induction of double strand breaks in the human genome. This would be particularly important for the competitive proliferation assays shown in Figure 6 A-B. Moreover, it is stated in the figure legend that sgRPS19 was used as a positive control for an essential gene, but these dates are not shown in Figure 6A and 6B. Also, as these two panels aim to identify the effect of USF2 as a regulator of SEM cell survival, it would be necessary to compare it to the effect of HOXA9 disruption by including sgRNAs targeting HOXA9. Finally, the authors should show a longer time course for the competitive competition assay than 12 days to determine attenuated effects on cell survival.

Thank you for the reviewer’s comments. To rule out the possibility that sgRNA-induced induction of double strand breaks may affect *HOXA9* expression and survival, we have checked the enrichment of several well tested sgRNAs in our CRISPR screen. Although double-strand breaks and indels were generated by these sgRNAs, none of them were enriched in our CRISPR screen nor shown to regulate *HOXA9*. We have now also repeated the competitive proliferation assay using sgRNA against ROSA26 endogenous gene, which demonstrated the same result to sgLuc. We apologize about the confusion of sgRPS19. The figure legend has been amended. We agree with the reviewer that comparable disruption of *HOXA9* in the proliferation assay should be included as a control. We have conducted the experiment and include the data in Figures 5C-65. Finally, we have extended the competitive proliferation assay to 23 days to determine attenuated effects on cell survival.

Figure 7 claims to validate the synergistic effect of combined inactivation of USF1 and USF2 in HOXA9 regulation. The authors should include a competitive proliferation assay to determine the effects of the single knockout of USF1 and USF2 vs the double knockout in SEM cells. Experiments should be validated in another MLLr human cell line. This assay should also include appropriate positive and negative controls, as mentioned above.

Thank you for the reviewer’s comments. We have repeated the competitive proliferation assay to determine the effects of the single knockout of USF1 and USF2 vs the double knockout in SEM cells and other MLLr leukemia cell lines (OCI-AML2 and MOLM13 with proper controls. We have added these data to Figures 5I-5J and Figure 5—figure supplement 1G.

Also, in Figure 7 D-E, it is necessary to include results from the single USF1 knockout.

Thank you for the reviewer’s comments. We have repeated these experiments using single knockout targeted with sgUSF1.

Legends to all supplementary figures are missing.

We apologize for the error. The supplemental figure legends are provided now.

Reviewer #3 (Significance):The authors have used state-of-the-art technology to generate a novel cell line to study the transcriptional regulation of HOXA9, a transcription factor with important implications in leukemogenesis. They identify USF2 as a new mechanism as to how HOXA9 may be regulated in leukemia cells. If they can address the main issues listed above, this work could be relevant for other scientists studying haematological malignancies. The work might also raise some interest in a broader audience due to the technical aspects of the manuscript.

Thank you for the reviewer’s positive comments.